# Improving Diversity in Language Models:
# When Temperature Fails, Change the Loss

**Alexandre Verine** [* 1]  **Florian Le Bronnec** [* 2 3]
**Kunhao Zheng** [2 4]  **Alexandre Allauzen** [2]  **Yann Chevaleyre** [2]  **Benjamin Negrevergne** [2]

## Abstract

Increasing diversity in language models is a challenging yet essential objective. A common approach is to raise the decoding temperature. In this work, we investigate this approach through a simplistic yet common case to provide insights into why decreasing temperature can improve quality (Precision), while increasing it often fails to boost coverage (Recall). Our analysis reveals that for a model to be effectively tunable through temperature adjustments, it must be trained toward coverage. To address this, we propose rethinking loss functions in language models by leveraging the Precision-Recall framework. Our results demonstrate that this approach achieves a substantially better trade-off between Precision and Recall than merely combining negative log-likelihood training with temperature scaling. These findings offer a pathway toward more versatile and robust language modeling techniques.

## 1. Introduction

Autoregressive language models (LMs) (Bengio et al., 2000) have demonstrated impressive capabilities in modeling natural language. By scaling both the data and the number of parameters, current transformer-based approaches (Touvron et al., 2023; DeepSeek-AI, 2024) are now able to produce expressive language models, capable of generating texts close to human-written ones. With these advances, evaluation has become a critical issue, and new metrics have been developed to better assess the generative abilities of these

models. Namely, several works have proposed to study two kind of errors these models can make (Pillutla et al., 2021; Shao et al., 2017; Le Bronnec et al., 2024):

1. The LM generates texts unlikely under the true distribution of language, leading to outputs of limited *quality*.
2. The LM focuses on producing a limited set of patterns or phrases, thereby reducing *diversity* in the outputs.

The analysis of these errors in generative models has been formalized through the Precision–Recall (P&R) framework, introduced by Sajjadi et al. (2018). *Precision* measures the proportion of generated samples that are plausible under a reference distribution, thereby assessing the *quality* of the model. *Recall* quantifies the proportion of the reference distribution that is covered by the generated samples, reflecting the model's *coverage*. Recall captures more than just sample diversity, it evaluates the diversity of samples that are likely under the reference distribution. This contrasts with metrics that operate independently of the reference distribution, such as those based on entropy (Theis et al., 2016; Friedman & Dieng, 2023) or vocabulary distinctiveness (Zhu et al., 2018), which may indicate high diversity even for random-like samples.

The P&R trade-off is crucial and varies by application domain. In code generation, high Precision is essential for producing runnable code, whereas in open-ended creative tasks, high Recall is key to generating diverse and engaging content. Independently of the P&R framework, the need to tune this trade-off has motivated the development of a variety of post-training corrective methods (Holtzman et al., 2020a). Among them, one of the simplest and most widely used methods is to adjust the decoding temperature (Fan et al., 2018; Zheng et al., 2024). Increasing the temperature directly increases the entropy i.e., the diversity of the model, however its impact on the Recall is not well understood. In this work, we aim to answer the following question:

**Question 1:** *What is the impact of adjusting the temperature of a LM in terms of Precision and Recall?*

We address this question by showing that temperature adjustments have a mixed impact on the P&R trade-off. While lowering the temperature can improve Precision, increasing

---

[*]Equal contribution  [1]École Normale Supérieure, Université PSL, DIENS, Paris, France [2]Miles, LAMSADE, Université Paris-Dauphine-PSL, Paris, France [3]Sorbonne Université, CNRS, ISIR, Paris, France [4]Meta FAIR, Paris, France. Correspondence to: Alexandre Verine <alexandre.verine@ens.fr>, Florian Le Bronnec <florian.le-bronnec@dauphine.psl.eu>.

*Proceedings of the 42nd International Conference on Machine Learning*, Vancouver, Canada. PMLR 267, 2025. Copyright 2025 by the author(s).

it often reduces Recall, contrary to the common intuition that higher temperature improves diversity. Based on this observation, we argue that achieving a better P&R trade-off through temperature scaling requires training the model to prioritize Recall. This naturally leads to the following question:

**Question 2:** *How can we train models for a higher Recall?*

We address Questions 1 and 2 by making the following contributions:

- **Impact of Temperature Scaling:** Section 4 presents Theorem 4.2, which analyzes the effect of temperature scaling on the P&R trade-off. We further refine this analysis in Proposition 4.3 by examining specific artificial cases.
- **Recall-Oriented Loss Functions:** In Section 5 we show that three existing loss functions fit in the P&R framework and optimize for Precision. We build upon this and introduce modified versions of these losses to enhance Recall.
- **Empirical Trade-off Evaluation:** In Section 6, we empirically validate our theoretical findings. Notably, we demonstrate that in most experiments, our proposed losses improve Recall and can lead to a model that achieves a better P&R trade-off through temperature scaling.

Our study suggests that focusing coverage at the training stage can produce language models that are more versatile with temperature scaling.

## 2. Background

### 2.1. Generative Models

In language models, we consider sequences of $L$ tokens, $\boldsymbol{x} = (x_1, \ldots, x_L) \in \mathcal{V}^L$ where $\mathcal{V}$ is the vocabulary set of cardinality $V$. We denote $\mathcal{P}(\mathcal{V}^L)$ as the set of all probability distributions over the sequence of $L$ tokens. The objective of training a generative model is to approximate the true data distribution $P$ by learning a parameterized distribution $Q_\theta \in \mathcal{P}(\mathcal{V}^L)$, modeled as a product of probabilities conditional to preceding tokens $\boldsymbol{x}_{<l} := (x_1, \ldots, x_{l-1})$:

$$Q_\theta(\boldsymbol{x}) = \prod_{l=1}^{L} Q_\theta(x_l \mid \boldsymbol{x}_{<l}), \tag{1}$$

where $Q_\theta(x_l \mid \boldsymbol{x}_{<l})$ denotes the conditional probability of token $x_l$ given the preceding tokens $\boldsymbol{x}_{<l}$, modeled as a softmax distribution over the output of a neural network $F_\theta$, parameterized by $\theta$. This network is trained to minimize the negative log-likelihood (NLL) of the data:

$$\mathcal{L}_{\mathrm{NLL}}(\theta) = -\mathbb{E}_{\boldsymbol{x} \sim P} \left[ \sum_{l=1}^{L} \log Q_\theta(x_l \mid \boldsymbol{x}_{<l}) \right]. \tag{2}$$

The NLL optimization problem is equivalent to minimizing $\mathcal{D}_{\mathrm{KL}}(P \| Q_\theta)$, the Kullback-Leibler divergence between the true distribution $P$ and the learned distribution $Q_\theta$.

**Notations:** For clarity, we omit the dependency on the parameters $\theta$ when it is clear from the context and use $Q$ instead of $Q_\theta$. When the parameters are fixed, meaning the gradient is detached, we denote this by $\bar{Q}$, implying that $\nabla_\theta \bar{Q} = 0$. Probabilities conditioned on the context $\boldsymbol{x}_{<l}$ are denoted as $P_{<l}(\cdot) = P(\cdot \mid \boldsymbol{x}_{<l})$ and $Q_{<l}(\cdot) = Q(\cdot \mid \boldsymbol{x}_{<l})$.

### 2.2. Temperature and sampling

Once the model has been trained, it can be used to generate new sequences by repeatedly sampling each token $x_l$ from $Q_{<l}$ using $\boldsymbol{x}_{<l}$ as the context (i.e., $x_l \sim Q(\cdot \mid \boldsymbol{x}_{<l})$). However, instead of sampling directly from $Q_{<l}$, practitioners generally introduce a new distribution $Q_{<l}^t$ which is based on $Q_{<l}$, but features an adjustable temperature parameter $t$ used to increase the entropy of $Q_{<l}^t$ and generate more diverse samples when necessary. For a given context $\boldsymbol{x}_{<l}$, the temperature-adjusted distribution $Q_{<l}^t$ is defined as:

$$Q_{<l}^t(x) = \frac{Q_{<l}(x)^{1/t}}{\sum_{x_i \in \mathcal{V}^L} Q_{<l}(x_i)^{1/t}}. \tag{3}$$

Note that setting the temperature $t = 1$ recovers the original distribution $Q_{<t}$. Setting the temperature to any value $t > 1$ increases the entropy of $Q_{<l}^t$ (compared to $Q_{<l}$), and $Q_{<l}^t$ becomes increasingly close to uniform distribution over $\mathcal{V}^L$ as $t \to \infty$. Conversely, setting the temperature to any value $t < 1$ decreases the entropy of $Q_{<l}^t$ leading to a deterministic distribution $Q_{<l}^t$ for $t = 0$.

### 2.3. Precision and Recall for Generative Models

In practice, increasing the temperature of the model distribution $Q^t$ increases the entropy of the generated samples, but it does not always lead to better coverage of the target distribution $P$. To properly assess the effect of temperature scaling, we adopt *Precision* and *Recall* (P&R) metrics for generative models. Inspired by classification, an intuitive definition of these metrics for generative models was introduced by Kynkäänniemi et al. (2019), based on comparing the supports of the two distributions, as follows.

**Definition 2.1** (Precision and Recall - (Kynkäänniemi et al., 2019)). Let $P, Q \in \mathcal{P}(\mathcal{V}^L)$. The *Precision* $\bar{\alpha}$ and *Recall* $\bar{\beta}$ of $Q$ with respect to $P$ are defined as:

$$\bar{\alpha} = Q(\mathrm{Supp}(P)) \quad \text{and} \quad \bar{\beta} = P(\mathrm{Supp}(Q)). \tag{4}$$

Computing these metrics requires estimating the supports of the distributions, which can be achieved through sampling followed by $k$-nearest neighbors support estimation. This approach is practical and has been widely used to assess the quality and coverage of both image generative models and language models (Kynkäänniemi et al., 2019; DeVries et al., 2020; Song et al., 2023; Le Bronnec et al., 2024).

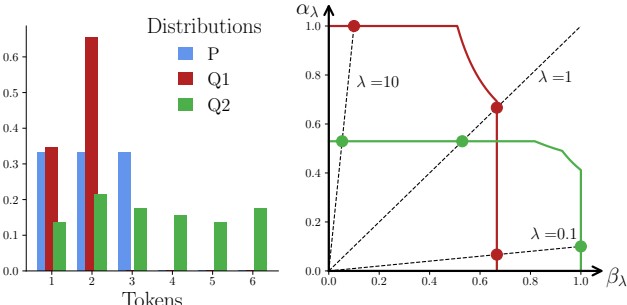

Figure 1: Examples of PR-Curves. $Q_1$ has a higher Precision than $Q_2$ but a lower Recall. Values of $(\alpha_\lambda, \beta_\lambda)$ are plotted for $\lambda \in \{0.1, 1, 10\}$.

While convenient to compute, these metrics have theoretical limitations, particularly in the case of tempered distributions, where adjusting the temperature does not alter the support of the model distribution. To alleviate these limitations, Sajjadi et al. (2018) have introduced an extension of Precision and Recall that compares the densities of $P$ and $Q$ instead of simply comparing their support. To achieve this, they introduce a new parameter $\lambda \in [0, \infty]$, which helps identify regions where $Q$ is at least $\lambda$ times denser than $P$. When $\lambda$ is high, the focus is on regions where $Q$ assigns probability mass while $P$ does not, which characterizes loss of Precision. Conversely, when $\lambda$ is low, the focus shifts to regions where $Q$ assigns very little mass while $P$ does, which reduces the Recall. The parameter $\lambda$ thus controls the trade-off between Precision and Recall, allowing the definition of the PR-Curve.

**Definition 2.2** (PR-Curve—(Sajjadi et al., 2018))**.** Let $P, Q \in \mathcal{P}(\mathcal{V}^L)$. The *PR-Curve* is the set $\mathrm{PRD}(P, Q)$ defined as:

$$\mathrm{PRD}(P, Q) = \{(\alpha_\lambda, \beta_\lambda) \mid \lambda \in [0, \infty]\}, \quad (5)$$

where:

$$\alpha_\lambda(P \| Q) = \sum_{\boldsymbol{x} \in \mathcal{V}^L} \min(\lambda P(\boldsymbol{x}), Q(\boldsymbol{x})), \quad (6)$$

$$\text{and} \quad \beta_\lambda(P \| Q) = \sum_{\boldsymbol{x} \in \mathcal{V}^L} \min(P(\boldsymbol{x}), Q(\boldsymbol{x})/\lambda). \quad (7)$$

The two definitions are related: $\alpha_\infty = \bar{\alpha}$ and $\beta_0 = \bar{\beta}$. The values of $\alpha_\lambda$ for high $\lambda$ captures the *quality*, while the values of $\beta_\lambda$ for low $\lambda$ captures the *diversity* with respect to the true distribution. Figure 1 shows examples of PR curves.

## 3. Related Works

With the rise of models exhibiting remarkable generative capabilities and the need to assess both their fidelity and diversity, several works have demonstrated the versatility of P&R metrics for developing and evaluating generative models (Sajjadi et al., 2018; Song et al., 2023). To balance

this trade-off, some studies have explored modifications to the sampling process (Brock et al., 2019). In the case of language models, adjusting the decoding process is a common strategy (Fan et al., 2018), with some methods explicitly targeting improved diversity (Chang et al., 2023). We focus specifically on the temperature parameter, whose limitations have been highlighted in multiple studies (Peeperkorn et al., 2024; Holtzman et al., 2020b), mainly from an empirical perspective.

Beyond inference, various training methods have been developed to address this trade-off. For instance, $f$-divergence minimization has shown promising results in image generation (Nowozin et al., 2016; Grover et al., 2018), with Verine et al. (2023) explicitly framing their approach within the P&R paradigm. In text generation, several studies have examined the role of different $f$-divergences in RLHF (Wang et al., 2023; Go et al., 2023; Sun & Schaar, 2024), either for reward modeling or regularization. For autoregressive models, various works have proposed training with modified loss functions (Ji et al., 2023; Kang & Hashimoto, 2020; Pang & He, 2021). While introduced with different motivations, we later unify these methods within the P&R framework and show that they all effectively maximize surrogates of Precision.

## 4. Temperature and PR-Curves

In this section, we analyze the impact of temperature on P&R. First we express a general bound on Precision and Recall as a function of temperature. Then, in order to gain deeper insights, we craft a simplified realistic distribution, and characterize how Precision and Recall change as we increase the temperature of this distribution.

**General case:** Our first result relies on the concept of *sparsity* defined as follows.

**Definition 4.1** (Sparsity of a distribution)**.** Given a context $\boldsymbol{x}_{<l} \in \mathcal{V}^{l-1}$ the *sparsity* of a distribution $P_{<l}$ is defined as $|\mathrm{Supp}(P_{<l})|/V$. We say that a distribution is *sparse* when $|\mathrm{Supp}(P_{<l})|/V \ll 1$. More generally, the *sparsity* of a distribution $P$ over sequences in $\mathcal{V}^L$ is defined as $|\mathrm{Supp}(P)|/V^L$.

**Theorem 4.2** (Impact of sparsity on P&R)**.** *Let* $P, Q_\theta \in \mathcal{P}(\mathcal{V}^L)$*, then for any temperature* $t$*, we have:*

$$\alpha_\lambda(P \| Q_\theta^t) \le \frac{|\mathrm{Supp}(P)|}{V^L} e^{ZL/t} \quad (8)$$

$$\text{and} \quad \beta_\lambda(P \| Q_\theta^t) \le \frac{1}{\lambda} \frac{|\mathrm{Supp}(P)|}{V^L} e^{ZL/t}, \quad (9)$$

*where* $Z = \max_{\boldsymbol{x} \in \mathcal{X}_V^K, \, l \in \{1, \dots L\}, \, i,j \in \mathcal{V}} F_\theta(\boldsymbol{x}_{<l})_i - F_\theta(\boldsymbol{x}_{<l})_j$ *the highest difference between the logits of the model.*

The proof of this theorem is provided in Appendix A.1. We observe that the sparser the target distribution, the harder it is for $Q_\theta$ to achieve good Precision and Recall.

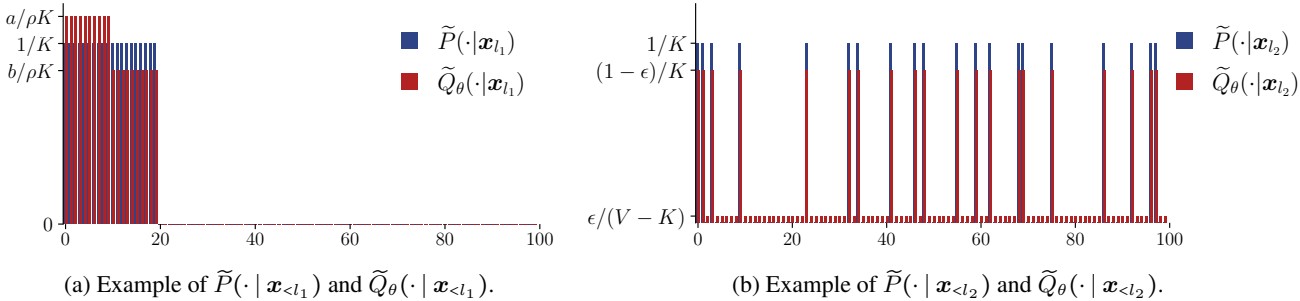

Figure 2: Example $\widetilde{P}$ and $\widetilde{Q}_\theta$ for $V = 100$, $K = 20$ and $a/\rho = 1.45$, and $\epsilon = 0.15$. $K/V$ represent the sparsity of the target distribution. $\rho$ is the proportion of tokens underrepresented at the level $b/\rho$ in $Q_\theta$ at $l_1$. $\epsilon$ is the noise level at $l_2$.

In practice, the true distribution $P_{<l}$ is sparse for most contexts because of the number of grammatical, syntactic, or semantic rules that apply and force $P_{<l}$ to be null for most tokens. In contrast, the model distribution $Q_{<l}$ is not sparse due to the softmax normalization, thus it will assign nonzero probabilities even for unacceptable tokens. Intuitively, increasing the temperature will increase the probability of these tokens and introduce a disproportionate number of unacceptable generations, effectively unlearning the language rules, and reducing both Precision and Recall. We empirically estimate the sparsity of reference distributions in Section 6.

**Artificial Case Analysis:** To gain deeper insight on the impact of increasing the model temperature on the PR-Curve, we analyze a specific case. We choose a target distribution $\widetilde{P}$ where all $\widetilde{P}_{<i}$ are sparse uniform distributions over small subset of $K$ tokens, and a model distribution $\widetilde{Q}_\theta$ that matches $\widetilde{P}$ everywhere except at the two specific positions in the sequence $l_1$ and $l_2$, i.e, $\forall \boldsymbol{x} \in \mathcal{V}^L$, $\forall i \notin \{l_1, l_2\}$, $\widetilde{Q}_\theta(\cdot \mid \boldsymbol{x}_{<i}) = \widetilde{P}(\cdot \mid \boldsymbol{x}_{<i})$. For position $l_1$ and $l_2$, the conditional distributions are defined as follows:

$$\widetilde{Q}_\theta(x \mid \boldsymbol{x}_{<l_1}) = \begin{cases} \frac{a}{\rho K}, & \text{if } x < \rho K, \\ \frac{b}{\rho K}, & \text{if } \rho K \leq x \leq K, \\ 0, & \text{otherwise,} \end{cases} \quad (10)$$

$$\widetilde{Q}_\theta(x \mid \boldsymbol{x}_{<l_2}) = \begin{cases} \frac{1-\epsilon}{K}, & \text{if } \widetilde{P}_{<l_2}(x) \neq 0, \\ \frac{\epsilon}{V-K}, & \text{otherwise.} \end{cases} \quad (11)$$

This is illustrated in Fig. 2 (a) and (b) respectively. This simple setting allows a full characterization of the PR-Curve across different temperatures while remaining theoretically meaningful. Notably, we can show that the inequalities (8) and (9) are tight for high temperatures $t \geq Z + \log(K) - \log(\lambda)/L$, demonstrating that $\widetilde{Q}_\theta$ achieves the optimal Precision-Recall tradeoff in this regime. The exact expressions for the PR-Curve and its rate of change are provided in Appendix A, Prop. A.2 and A.4. Connections to real-world behavior are discussed in Section 6. The results can be summarized in the following proposition:

**Proposition 4.3** (Informal—P&R with temperature). *Let $\widetilde{P}, \widetilde{Q}_\theta \in \mathcal{P}(\mathcal{V}^L)$ as described in the previous paragraph.*

- *For high values of $\lambda$, the Precision $\alpha_\lambda$ decreases as the temperature $t$ increases.*

- *For low values of $\lambda$, the Recall $\beta_\lambda$ decreases for temperatures $t \geq t_0$, provided that any one of the following mild conditions is met:*

  - *the vocabulary size $V$ is much larger than the target support size $K$ ($K \ll V$),*
  - *$\widetilde{Q}_\theta$ is good approximation of $\widetilde{P}$, (i.e., $a \approx 1$, $b \approx 1$, $\rho \approx 1$, $\epsilon \ll 1$).*

Further details about $\widetilde{Q}_\theta$ The formal theorem, its proof and details on $t_0$ are available in Appendix A.2.

For Recall, a common behavior is an initial increase with rising temperature, peaking at some value $t_0$. Beyond this point, Recall tends to decline and eventually converges to a level that is consistently lower than the value at $t = 1$. To obtain a model capable of achieving both higher Precision and higher Recall through temperature scaling, it is beneficial to train the model with a strong emphasis on Recall. Precision may then be improved by lowering the temperature. This motivates the objective of the next section: developing training strategies that produce a diverse set of models.

## 5. Training model for diverse output

Training models for specific Precision-Recall trade-offs has been explored in image generation, notably by minimizing alternative $f$-divergences (Verine et al., 2023). However, these methods depend on assumptions that are incompatible with the causal structure of language generation, rendering them unsuitable for direct use in language models. Instead, existing methods adapt the training loss to weight samples differently from the standard negative log-likelihood (NLL) loss. In this section, we focus on three representative methods: *Trunc* (Kang & Hashimoto, 2020), *GOLD* (Pang &

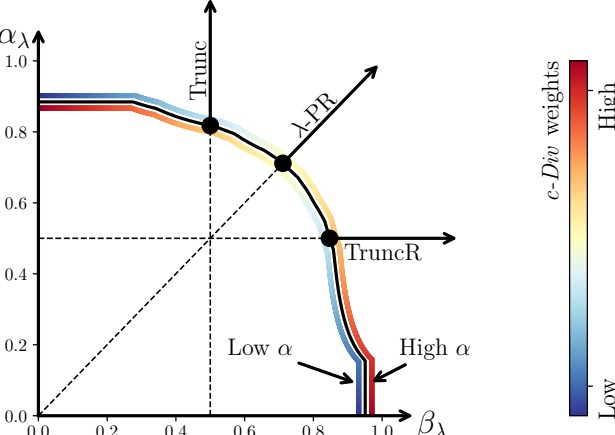

Figure 3: Illustration of trade-offs targeted by each method over the model's PR curve (black). The highlighted parts represents regions where two *c-Div* loss functions of opposite effects place more emphasis, red indicates high emphasis, blue low. *Trunc* and *TruncR* target a specific point on the PR-Curve, either maximizing it vertically (Precision) or horizontally (Recall), while $\lambda$-PR focuses on improving a point along the $y = \lambda x$ line.

He, 2021), and *TaiLr* (Ji et al., 2023). Although originally introduced with different motivations, we show that these methods consistently prioritize Precision over Recall. To address this imbalance, we propose alternative loss functions designed to enhance Recall. All theoretical proofs are provided in Appendix B. In the next section, we experimentally evaluate which methods are most effective in making models (1) more diverse and, most importantly, (2) more versatile.

All losses discussed in this section can be viewed as reweighted variants of the NLL, with weight computations detailed in Appendix C. Figure 3 illustrates the trade-offs each method targets along the model's PR curve.

### 5.1. Baseline Methods

***Trunc*** **by Kang & Hashimoto (2020).** This approach defines a target distribution $P^{\text{trunc}}$, obtained by renormalizing the original distribution $P$ over a subset $\mathcal{X}_{\text{trunc}} \subset \mathcal{X}$, satisfying $P(\mathcal{X}_{\text{trunc}}) = 1 - \Delta$ for a given constant $\Delta \in [0, 1]$. The method minimizes the NLL between this truncated distribution and the model distribution, resulting in a tighter upper bound on the total variation distance. In practice, the model is trained exclusively on samples with the highest log-likelihood values. This is done by selecting samples within the top $1 - \Delta$ quantile of the log-likelihood distribution, i.e., dynamically identifying the threshold $\delta \in \mathbb{R}$ such that $\mathbb{E}_{\boldsymbol{x} \sim P}\left[\mathbb{1}_{\{\bar{Q}_\theta(\boldsymbol{x}) \geq \delta\}}\right] = 1 - \Delta$ and then minimizing the

following loss:

$$\mathcal{L}_{\text{Trunc}}^{\Delta}(\theta) = -\mathbb{E}_{\boldsymbol{x} \sim P}\left[\sum_{l=1}^{L} \mathbb{1}_{\{\bar{Q}(\boldsymbol{x}) \geq \delta\}} \log Q_{<l}(x_l)\right]. \quad (12)$$

***GOLD*** **by Pang & He (2021).** *GOLD* (Generation by Off-policy Learning from Demonstrations) is a method that leverages off-policy learning to improve the quality of generated samples. One specific variant of this method, GOLD-$\delta$, has been shown to be equivalent to reweighting the gradients of the training samples' log-likelihood, thus further promoting highly probable tokens (Li et al., 2022). The authors propose the following loss:

$$\mathcal{L}_{\text{GOLD}}(\theta) = -\mathbb{E}_{\boldsymbol{x} \sim P}\left[\sum_{l=1}^{L} \bar{Q}_{<l}(x_l)^{\frac{1}{2}} \log Q_{<l}(x_l)\right]. \quad (13)$$

***TaiLr*** **by Ji et al. (2023).** *TaiLr* (Total Variation Guided Language Generation) is a method that aims to minimize the TV distance between the model distribution and the target distribution. To do so, the authors first propose an upper bound on the TV distance based the total variation of the conditional distributions. They then approximate the unknown conditional target distribution $P_{<l}$ as a mixture of the model's own distribution and a one-hot distribution:

**Definition 5.1** ($\gamma$-proxy distribution)**.** Let $x_l \sim P_{<l}$ be a token sampled from the conditional target distribution. Given $\gamma \in [0, 1]$, the $\gamma$-proxy distribution is defined as:

$$\widehat{P}_{<l}^{x_l}(\cdot) = \gamma \mathbb{1}_{\{x_k = \cdot\}} + (1 - \gamma) Q_{<l}(\cdot). \quad (14)$$

This $\gamma$-proxy is used to derive the following loss:

$$\mathcal{L}_{\text{TaiLr}}^{\gamma}(\theta) = -\mathbb{E}_{\boldsymbol{x} \sim P}\left[\sum_{l=1}^{L} \frac{\bar{Q}_{<l}(x_l)}{\gamma + (1 - \gamma)\bar{Q}_{<l}(x_l)} \log Q_{<l}(x_l)\right]. \quad (15)$$

### 5.2. From *Trunc* to *TruncR*

The *Trunc* method considers only a subset of the target distribution, meaning it does not penalize the model for missing samples outside this subset. As a result, it inherently allows for a loss of Recall. More precisely:

**Proposition 5.2** (*Trunc* optimizes Precision at a given Recall)**.** *Optimizing $\theta$ using the $\mathcal{L}_{\text{Trunc}}^{\Delta}$ loss is equivalent to optimizing Precision for a fixed value of Recall $\beta = 1 - \Delta$.*

This result is derived by minimizing the NLL between the truncated $P$ and $Q$. To reverse the trade-off, one might consider inverting $P$ and $Q$ in the loss. However, this approach is infeasible as it requires sampling from $Q$ and evaluating $P(\cdot|\boldsymbol{x}_{<l})$. To address this, we introduce a loss function that approximates minimizing the NLL between $P$ and $Q^{\text{trunc}}$:

$$\mathcal{L}_{\text{TruncR}}^{1-\Delta}(\theta) = -\mathbb{E}_{\boldsymbol{x} \sim P}\left[\sum_{l=1}^{L} \mathbb{1}_{\{\bar{Q}_\theta(\boldsymbol{x}) \leq \delta\}} \log Q_{<l}(x_l)\right] \quad (16)$$

such that $\mathbb{E}_{\boldsymbol{x}\sim Q}\left[\mathbb{1}_{\{\bar{Q}_\theta(\boldsymbol{x})\leq\delta\}}\right] = 1 - \Delta$. Note that this approach requires the same quantile approach to estimate the threshold $\delta$. We can show that this proposed loss achieves the opposite effect of the *Trunc* method:

**Proposition 5.3** (*TruncR* optimizes Recall at a given Precision). *Optimizing $\theta$ using $\mathcal{L}_{\mathrm{TruncR}}^{1-\Delta}$ is equivalent to optimizing Recall for a fixed value of Precision $\alpha = 1 - \Delta$.*

Training for a limited subset of the model distribution is not a new idea. In fact, Verine et al. (2024) train image generative models on limited subset of generated samples and also observe that this leads to increased Recall.

### 5.3. From *GOLD* to *c-Div*

By reweighting the log-likelihood of training samples, the *GOLD* method increases the likelihood of highly probable tokens. However, this reweighting can be generalized:

**Theorem 5.4** (Conditional Tsallis $\alpha$-Divergence Minimization). *Using Definition 5.1 with $\gamma = 1$, minimizing the conditional $\alpha$-divergence between $\widehat{P}_{<l}^{x_l}$ and $Q_{<l}$ for all $l \in \{1, \ldots, L\}$ and for all $\boldsymbol{x} \sim P$ is equivalent to minimizing*

$$\mathcal{L}_{\mathrm{cDiv}}^{\alpha}(\theta) = -\mathbb{E}_{\boldsymbol{x}\sim P}\left[\sum_{l=1}^{L}\bar{Q}_{<l}(x_l)^{1-\alpha}\log Q_{<l}(x_l)\right]. \quad (17)$$

In particular, *GOLD* minimizes $\mathcal{L}_{\mathrm{cDiv}}^{\alpha}(\theta)$ with $\alpha = 1/2$. It is well established that adjusting the $\alpha$ parameter in the minimization of $\alpha$-divergence significantly influences the trade-off between quality and diversity (Minka, 2005; Labeau & Cohen, 2019; Go et al., 2023). The relationship between $\alpha$-divergence and the PR-Curve has been analyzed in Verine (2024). Higher values of $\alpha$ lead to divergences that are more *mass-covering*, favoring Recall. In particular, optimizing a divergence with $\alpha > 1$ results in a more Recall-oriented approach compared to NLL.

### 5.4. From *TaiLr* to $\lambda$-PR

The *TaiLr* method aims to optimize the Total Variation (TV) distance, which corresponds to the point $1 - \alpha_\lambda(P\|Q)/2$ for $\lambda = 1$. However, due to practical constraints, the proposed loss is not a proper loss function and does not directly optimize the TV distance. Instead, it solves a different problem:

**Proposition 5.5.** *The optimal distribution $Q_{\theta^*}$ using the TaiLr method, i.e., $\mathcal{L}_{\mathrm{TaiLr}}^{\gamma}(\theta)$ with $\gamma > 0$, is given for every context $\boldsymbol{x}_{<l} \in \mathcal{V}^{l-1}$ by:*

$$\forall x \in \mathcal{V}, \quad Q_{\theta^*}(x \mid \boldsymbol{x}_{<l}) = \frac{P_{<l}(x)(1 - \gamma + V\gamma) - \gamma}{1 - \gamma}. \quad (18)$$

*In other words, $Q_{<l}(x) > P_{<l}(x)$ if $P_{<l}(x) > 1/V$, and $Q_{<l}(x) \leq P_{<l}(x)$ otherwise.*

The loss $\mathcal{L}_{\mathrm{TaiLr}}^{\gamma}$ is not a proper loss since its optimal distribution is not $P$. However, it effectively reduces the likelihood

of less probable tokens, likely increasing Precision. We propose to generalize this reasoning and propose the following non-proper loss function:

$$\mathcal{L}_{\lambda-\mathrm{PR}}^{\gamma}(\theta) = -\mathbb{E}_{\boldsymbol{x}\sim P}\left[\sum_{l=1}^{L}w(l,\lambda,\gamma)\log Q_{<l}(x_l)\right], \quad (19)$$

where

$$w(l,\lambda,\gamma) = \lambda^{\frac{l-1}{L}}\mathbb{1}_{\left\{\bar{Q}_{<l}(x_l)\leq\delta_{\lambda^{1/L}}\right\}}\frac{\bar{Q}_{<l}(x_l)}{\gamma + (1-\gamma)\bar{Q}_{<l}(x_l)},$$

and $\delta_{\lambda^{1/L}} = \frac{\lambda^{1/L}\gamma}{1-(1-\gamma)\lambda^{1/L}}$. We show that under the same assumptions than Ji et al. (2023), this loss can be used to optimize any point $(\alpha_\lambda, \beta_\lambda)$ on the PR-Curve for $\lambda \leq 1$:

**Theorem 5.6** ($\lambda$-*PR* optimizes the PR-Curve at $\lambda$). *Optimizing $\theta$ using the $\mathcal{L}_{\lambda-\mathrm{PR}}$ loss is equivalent to maximizing a lower bound on $(\alpha_\lambda, \beta_\lambda)$ on the PR-Curve for $\lambda \leq 1$.*

This loss generalizes $\mathcal{L}_{\mathrm{TaiLr}}^{\gamma}$ to any point on the PR-Curve for $\lambda \leq 1$. Notably, for $\lambda = 1$, the loss is equivalent to the *TaiLr* loss, as $\delta_\lambda = 1$ and $\lambda = 1$. For $\lambda < 1$, the $\lambda$-PR loss retains the effect described in Proposition 5.5 but additionally enforces the likelihood to remain below $\delta_{\lambda^{1/L}}$.

## 6. Experiments

In this section, we empirically assess the theoretical insights on temperature scaling (Section 4) and training methods (Section 5). Our experiments aim to address the following questions:

- To what extent do our simplified theoretical settings align with real-world language modeling scenarios?
- What is the impact of temperature scaling on the Precision-Recall trade-off in language models?
- How do the proposed training methods affect Recall?

To answer these questions, we conduct experiments on multiple tasks at different scales.

### 6.1. Evaluation Tasks and Metrics

We consider four tasks where quality and coverage can be measured in a meaningful way: two tasks of code generation, integer multiplication, and open ended generation. Full details regarding the different evaluation methods are detailed in Appendix D.2.

**CodeContests (Li et al., 2022).** We use the test set comprising 165 challenging problems. We propose to evaluate P&R using the widely used pass@$k$ metrics, (Chen et al., 2021), which is the expectation that at least one code sample is correct given a budget of $k$ samples. In this setup, we naturally consider **pass@1** as a proxy for **Precision**, since it measures exactly the portion of the outputs generated by

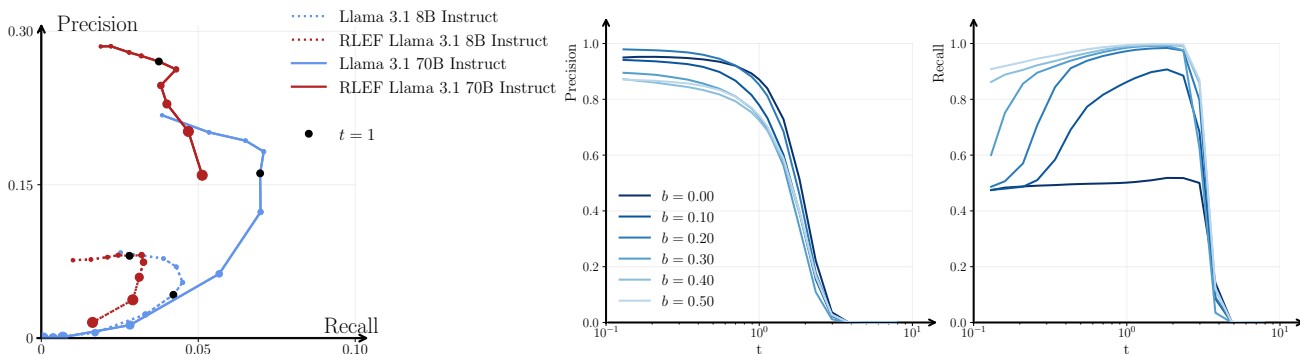

Figure 4: Effect of tuning $t$ between 0.2 (smaller dots) and 2 (larger dots) on P&R for Llama3.1 models on the CodeContests dataset.

Figure 5: Effect of the temperature on the P&R for the integers multiplication task, at different levels of underrepresentation $b$ in the training data. Recall increases with the temperature, then drops.

the model that are correct. Then, the boost from pass@1 to pass@$k$ depends largely on the diversity of the code samples, which is why we use **pass@100 - pass@1** as **Recall**.

**MathQA-Python (Chen et al., 2021).** We evaluate the P&R trade-off on the MathQA-Python dataset, which consists of simple Python code generation tasks. We use the same evaluation as in CodeContests, using pass@$k$ metrics.

**Integers multiplication.** The goal is to generate pairs of positive two-digit integers along with their product modulo 97, in the format $a_1a_2 \times b_1b_2 = c_1c_2$ (Papadopoulos et al., 2024). The reference distribution is uniform over the input integers and deterministic over the result. The model $Q_\theta$ is trained on synthetically sampled examples from $P$. Although this example is simple, we believe it models key characteristics of real-world patterns in natural language, where certain words exhibit a spread probability distribution or highly skewed distributions depending on the position.

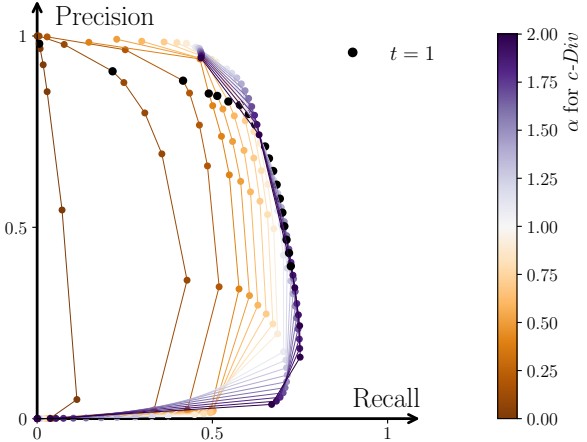

Figure 6: P&R for $t \in [0.1, 5]$ for the integer multiplication task for models trained with *cDiv* for different $\alpha$. By tuning $t$ on models with high Recall, we improve the P&R trade-off. The PR-Curves for high $\alpha$ dominate the curves for lower $\alpha$.

When training models over this dataset, we simulate underrepresented integers by adjusting their frequency in the training data, mimicking imperfect learning by $Q_\theta$. Specifically, the first digit of the first token is drawn with proportion $b$ from the first five digits and $1 - b$ from the last five. This setup closely matches the artificial cases, allowing us to validate their relevance empirically. As the reference distribution is known, we can compute the P&R metrics directly.

**Writing Prompts (Fan et al., 2018).** This task involves generating creative stories from given prompts. We analyze how different training losses and temperature settings impact the learned distribution $Q_\theta$ in generating text. Using the P&R metrics from Le Bronnec et al. (2024), we assess the trade-offs between fidelity and diversity in text generation.

### 6.2. Models

We use the following models in our experiments: **Llama3.1-8B/70B Instruct/RLEF** (Grattafiori et al., 2024) are general instruction-tuned models. These models are used for the CodeContests generation tasks and for the support sparsity estimation. **Olmo-1B** (Groeneveld et al., 2024), is a smaller pre-trained model. We finetune this model on the Writing-Prompts and MathQA-Python datasets using the proposed losses. **Llama3.2-3B** is finetuned on the WritingPrompts dataset. **Llama-Alpaca** is a Llama3.1-8B instruction tuned on the Alpaca dataset (Taori et al., 2023). This model is used to on WritingPrompts and MathQA-Python tasks. For Llama3.1-8B and Llama3.2-3B, we omit *Trunc* and *TruncR* results due to incompatibility of the original code with multi-GPU parallelism, which is required for training. Further training implementations are detailed in Appendix D.

### 6.3. Assumptions for the Theoretical Limits of P&R

**Sparsity of $P$.** In Theorem 4.2, we establish the limit of the PR-Curve as a function of the sparsity of $P$. To gain a practical insight on the actual sparsity of $P$, we compute an

Table 1: P&R of Olmo-1B trained on WritingPrompts dataset. Values higher than NLL are highlighted in bold. We report the MAUVE (Pillutla et al., 2021) score for reference.

| Method | MAUVE | P | R |
|---|---|---|---|
| NLL | 0.104 | 81.4 | 8.9 |
| *Trunc* ($\Delta = 0.25$) | 0.074 | **85.5** | 8.9 |
| *Trunc-R* ($\Delta = 0.25$) | 0.073 | **85.5** | **9.3** |
| *GOLD* ($\alpha = 0.5$) | 0.005 | **99.3** | 0.2 |
| *c-Div* ($\alpha = 1.4$) | 0.068 | 54.2 | **11.9** |
| *TaiLr* ($\lambda = 1, \gamma = 10^{-5}$) | 0.087 | **83.9** | **9.3** |
| $\lambda$-PR ($\lambda = 0.1, \gamma = 10^{-5}$) | 0.096 | 81.3 | **12.6** |

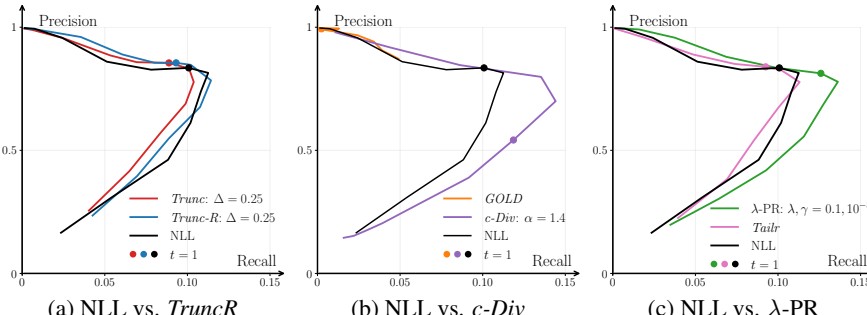

(a) NLL vs. *TruncR*  (b) NLL vs. *c-Div*  (c) NLL vs. $\lambda$-PR

Figure 7: P&R trade-off for $t \in [0.1, 1.8]$ for various training methods on the WritingPrompts dataset with Olmo-1B. Existing losses such as *Trunc, GOLD* and *TaiLr* tend to favor Precision over Recall. In contrast, our proposed losses improve Recall while maintaining comparable Precision when adjusting the temperature.

estimate of the sparsity of the distribution for two datasets: CodeContests and WritingPrompts. Our estimation is based number of unique tokens within the support of the conditional distribution $P(\cdot \mid \boldsymbol{x}_{<l})$. We approximate this by considering the truncated conditional distribution obtained from a strong pretrained model $P_\theta$, specifically counting the tokens that constitute the top-$p$ probability mass of $P_\theta(\cdot \mid \boldsymbol{x}_{<l})$. Full details are provided in Appendix D.2. Our experiments reveal significant sparsity in the conditional distributions relative to the vocabulary size $V$. Specifically, for CodeContests, the estimated support size is less than $0.1\%$ of $V$ when $p = 0.9$, and remains under $8\%$ when $p = 0.99$. These observations confirm the high sparsity of the target distribution, empirically validating that the PR-Curve of $Q_\theta$ deteriorates significantly when $t \gg 1$.

**Connection with artificial case.** These observations suggest that the artificial case in Theorem 4.2 captures features relevant to practice: the target distribution $P$ is highly sparse, and the pretrained model, viewed as $Q_\theta$, concentrates mass on a few tokens, with low residual spread across the rest, reflecting the theoretical structure.

### 6.4. Effect of Temperature on P&R

Results are shown in Figures 4, 5, 6 and 7. Results with other decoding methods are discussed in Appendix D.3.

Table 2: Sparsity of the target distribution (approximated upper bound on $|\mathrm{Supp}(P)|/V$ for various top-$p$ truncation thresholds of the approximate distribution, on the CodeContests and WritingPrompts datasets.

| Task \ $p$ | 0.9 | 0.95 | 0.99 |
|---|---|---|---|
| CodeContests | 0.03% | 0.08% | 8.08% |
| WritingPrompts | 3.86% | 7.80% | 24.9% |

**Lowering the temperature improves Precision but reduces Recall.** In Figure 4, Llama3.1-8B RLEF is the only set-up when the temperature has no effect on the Precision. Across all other tasks, decreasing the temperature ($t < 1$) leads to higher Precision, but always at the cost of lower Recall. This aligns with our analysis in Section 4.

**Increasing the temperature has a limited or negative impact on Recall.**

As the temperature increases, we observe the expected behavior in the toy multiplicative task: Recall initially improves, but beyond a certain threshold, it begins to decline and eventually drops to zero. In the more complex WritingPrompts and CodeContests tasks, this effect is less pronounced: Recall exhibits little or no improvement before ultimately decreasing. These observations are consistent with the theoretical findings in Section 4, and in particular with Proposition 4.3, which shows that while higher temperatures may temporarily enhance Recall, they eventually cause it to decline to zero..

### 6.5. Training for Recall

Figure 8 presents the P&R of the integer multiplication task for different training methods at different parameter settings. Table 1 and 3 presents the P&R of various models trained with different losses on the WritingPrompts and MathQA dataset. As predicted, **baseline losses favor Precision** at the expense of Recall, whereas **our proposed losses successfully improve Recall**. This is consistent with our theoretical analysis in Section 5.

### 6.6. Enhancing the Temperature P&R Trade-off

**Recall-optimal point.** On all tasks, when the baseline NLL is temperature-tuned to maximize Recall, we can tune our methods to reach a superior Recall level with the same Precision. This is verified on Figure 6 for integers multipli-

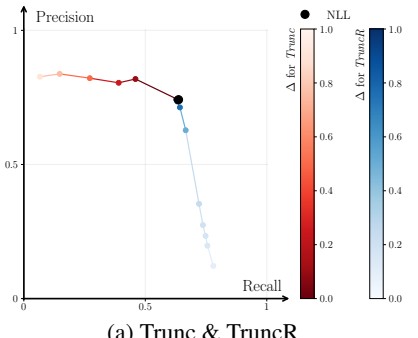 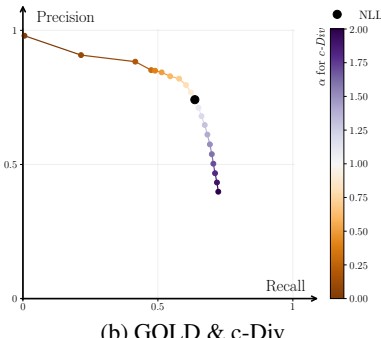 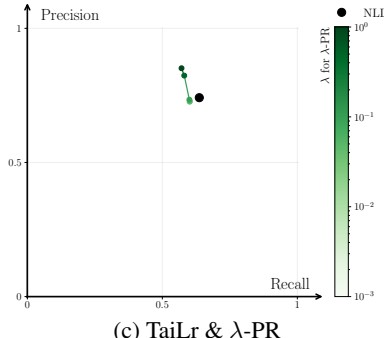

| (a) Trunc & TruncR | (b) GOLD & c-Div | (c) TaiLr & $\lambda$-PR |

Figure 8: P&R trade-off at $t = 1$ with the proposed losses on the integer multiplication task. Baselines *Trunc*, *GOLD*, and *TaiLr* improve Precision. In contrast, our proposed losses improve Recall.

cation, on Figure 7 for the WritingPrompts dataset. This is also verified on MathQA dataset on Table 4a, where we report *c-Div*-trained models achieving a higher Recall than NLL at the same Precision level.

**Precision-optimal point.** We identify scenarios in which our approach consistently achieves higher Recall across the entire spectrum of Precision levels attainable by NLL. This is demonstrated on Figures 6 and 7b, and for Llama-Alpaca, with the *c-Div* loss in Table 4b.

Figures 6 and 7 illustrate the evolution of P&R as temperature varies. Beyond improving Recall at $t = 1$, **our proposed losses achieve a better overall P&R trade-off**. Notably, they allow for slightly better Precision at the same Recall level as NLL by adjusting the temperature, but also allow for a significantly better Recall at the same Precision level. This suggests that training for Recall can make tem-

perature scaling more effective in balancing P&R. However, we observe that these loss functions can be less stable than NLL in practice.

# 7. Conclusion

Our study highlights the limitations of temperature scaling as a mean to improve diversity in language models. While lowering the temperature enhances Precision, increasing it often fails to significantly boost Recall. Through theoretical analysis and empirical evidence, we show that models must be trained explicitly for Recall to make temperature tuning more effective. To this end, we propose alternative loss functions that achieve a better trade-off between Precision and Recall. Our results suggest that refining training objectives is a more effective approach than relying solely on decoding strategies, and can make models more versatile.

Table 3: Comparison of training methods on the WritingPrompts on MathQA-Python datasets. All generations were sampled with temperature $t = 1$, except for Llama-Alpaca, where a lower temperature $t = 0.5$ was used to mitigate degenerate outputs.

| Task | | WritingPrompts | | | | | | | MathQA-Python | | | | | |
|---|---|---|---|---|---|---|---|---|---|---|---|---|---|---|
| Model | | Olmo-1B | | Llama-Alpaca | | Llama3.2-3B | | | Olmo-1B | | | Llama-Alpaca | | |
| Method | | P | R | | P | R | | P | R | | P | R | | P | R |
| NLL | - | 81.4 | 8.9 | - | **83.3** | 4.4 | - | **77.4** | 8.1 | - | **42.0** | 36.6 | - | **8.8** | 39.0 |
| *Trunc-R* | $\Delta = 0.25$ | **85.5** | 9.3 | - | - | - | - | - | - | $\Delta = 0.1$ | 29.8 | 43.3 | - | - | - |
| *c-Div* | $\alpha = 1.4$ | 54.2 | 11.9 | $\alpha = 1.4$ | 82.2 | 12.6 | $\alpha = 1.3$ | 72.5 | 17.5 | $\alpha = 1.4$ | 30.1 | 46.1 | $\alpha = 1.4$ | 8.2 | **43.4** |
| $\lambda$-PR | $\lambda = 0.1$ $\gamma = 10^{-5}$ | 81.3 | **12.6** | $\lambda = 0.5$ $\gamma = 10^{-7}$ | 57.1 | **26.9** | $\lambda = 0.9$ $\gamma = 10^{-5}$ | 59.9 | **19.0** | $\lambda = 0.1$ $\gamma = 10^{-7}$ | 6.4 | **48.1** | $\lambda = 0.1$ $\gamma = 10^{-5}$ | 8.3 | 42.1 |

Table 4: *c-Div* trained models achieving better tradeoffs than NLL on MathQA-Python.

(a) Achieving higher Precision than NLL at highest Recall

| Model | | Olmo-1B | | Llama-Alpaca | | |
|---|---|---|---|---|---|---|
| Method | | P | R | | P | R |
| NLL | $t = 1.6$ | 0.20 | 0.47 | $t = 1$ | 0.067 | 0.49 |
| *c-Div* | $\alpha = 1.4, t = 1$ | 0.21 | 0.50 | $\alpha = 1.4, t = 0.8$ | 0.087 | 0.49 |

(b) Achieving higher Recall than NLL at highest Precision

| Model | | Llama-Alpaca | |
|---|---|---|---|
| Method | | P | R |
| NLL | $t = 1.6$ | 0.10 | 0.10 |
| *c-Div* | $\alpha = 1.4, t = 1$ | 0.10 | 0.14 |

## Acknowledgements

This work has been partly funded through project ACDC ANR-21-CE23-0007. This work was granted access to the HPC resources of IDRIS under the allocations 2025-A0181016159, 2025-AD011014053R2, 2025-A0171014638, 2025-AD011014022R2 made by GENCI.

## Impact Statement

This paper presents work whose goal is to advance fundamental algorithmic development. There are many potential societal consequences of our work, none which we feel must be specifically highlighted here.

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

## A. Proof of results in Section 4

### A.1. Proof of Theorem 4.2

For improved clarity and readability, we introduce the concept of *acceptable tokens*.

**Definition A.1** (Set of acceptable tokens). Given a context $\boldsymbol{x}_{<l} \in \mathcal{V}^{l-1}$ the set of acceptable tokens is $\mathcal{S}(\boldsymbol{x}_{<l}) := \mathrm{Supp}(P_{<l})$, i.e., the set of tokens with non-zero probabilities in $P_{<l}$.

Let us first recall the theorem:

**Theorem.** *Let $P, Q_\theta \in \mathcal{P}(\mathcal{X}_V^K)$, then for any temperature $t$, we have:*

$$\alpha_\lambda(P\|Q_\theta^t) \leq \frac{|\mathrm{Supp}(P)|}{V^L} e^{ZL/t} \tag{20}$$

$$and \quad \beta_\lambda(P\|Q_\theta^t) \leq \frac{1}{\lambda} \frac{|\mathrm{Supp}(P)|}{V^L} e^{ZL/t}, \tag{21}$$

*where $Z = \max_{\boldsymbol{x} \in \mathcal{X}_V^K} \max_{l \in \{1, \dots L\}} \max_{i,j \in \mathcal{V}} F_\theta(\boldsymbol{x}_{<l})_i - F_\theta(\boldsymbol{x}_{<l})_j$ the highest difference between the logits of the model and $|\mathrm{Supp}(P)| = \sum_{x_1 \in \mathcal{S}(\varnothing)} \sum_{x_2 \in \mathcal{S}(x_1)} \cdots \sum_{x_L \in \mathcal{S}(\boldsymbol{x}_{<L})} 1$.*

*Proof.* Recall that $\beta_\lambda = \alpha_\lambda / \lambda$ therefore we will study the behavior of $\alpha_\lambda$ with $t$ and $\lambda$:

$$\alpha_\lambda(P\|Q_\theta^t) = \sum_{\boldsymbol{x} \in \mathcal{X}_V^L} \min\left(\lambda P(\boldsymbol{x}), Q_\theta^t(\boldsymbol{x})\right), \tag{22}$$

$$= \sum_{\boldsymbol{x} \in \mathcal{X}_V^L} \min\left(\lambda P(\boldsymbol{x}), \prod_{l=1}^L Q_\theta^t(x_l \mid \boldsymbol{x}_{<l})\right), \tag{23}$$

Since $P(\boldsymbol{x}) = 0$ if $\boldsymbol{x} \notin \mathrm{Supp}(P_\theta)$ by definition then we can restrict the sum to $\boldsymbol{x} \in \mathrm{Supp}(P_\theta)$:

$$\alpha_\lambda(P\|Q_\theta^t) = \sum_{\boldsymbol{x} \in \mathrm{Supp}(P_\theta)} \min\left(\lambda P(\boldsymbol{x}), \prod_{l=1}^L Q_\theta^t(x_l \mid \boldsymbol{x}_{<l})\right). \tag{24}$$

If we denote $\Delta_{l, \boldsymbol{x}_{<l}, j} = \max_i F_\theta(\boldsymbol{x}_{<l})_i - F_\theta(\boldsymbol{x}_{<l})_j$, we can write the conditional probability as:

$$Q_\theta^t(x_j \mid \boldsymbol{x}_{<l}) = \frac{\exp(F_\theta(x_j \mid \boldsymbol{x}_{<l})/t)}{\sum_{k=1}^V \exp(F_\theta(x_k \mid \boldsymbol{x}_{<l})/t)} \tag{25}$$

$$= \frac{\exp(-\Delta_{l, \boldsymbol{x}_{<l}, j}/t)}{\sum_{k=1}^V \exp(-\Delta_{l, \boldsymbol{x}_{<l}, k}/t)}. \tag{26}$$

Thus, we can upper-bound the conditional probability by the probability for the most probable token:

$$Q_\theta^t(x_j \mid \boldsymbol{x}_{<l}) \leq \frac{1}{\sum_{k=1}^V \exp(-\Delta_{l, \boldsymbol{x}_{<l}, k})/t}. \tag{27}$$

We denote $Z = \max_{\boldsymbol{x} \in \mathcal{X}_V^K} \max_{l \in \{1, \dots L\}} \max_{i,j \in \mathcal{V}} F_\theta(\boldsymbol{x}_{<l})_i - F_\theta(\boldsymbol{x}_{<l})_j$ the highest difference between the logits of the model on the target distribution. We can then upper-bound the conditional probability by:

$$Q_\theta^t(x_j \mid \boldsymbol{x}_{<l}) \leq \frac{1}{\sum_{k=1}^V \exp(-Z/t)} = \frac{1}{V \exp(-Z/t)} = \frac{\exp(Z/t)}{V}. \tag{28}$$

Therefore, we can upper-bound the Precision by:

$$\alpha_\lambda(P\|Q_\theta^t) \le \sum_{\boldsymbol{x}\in\mathrm{Supp}(P)} \min\left(\lambda P(\boldsymbol{x}), \prod_{l=1}^{L} \frac{\exp(Z/t)}{V}\right) \tag{29}$$

$$= \sum_{\boldsymbol{x}\in\mathrm{Supp}(P)} \min\left(\lambda P(\boldsymbol{x}), \left(\frac{\exp(Z/t)}{V}\right)^L\right) \tag{30}$$

$$\le \sum_{\boldsymbol{x}\in\mathrm{Supp}(P)} \frac{\exp(ZL/t)}{V^L} \tag{31}$$

$$= |\mathrm{Supp}(P)| \frac{\exp(ZL/t)}{V^L}, \tag{32}$$

which concludes the proof for the Precision. $\qquad\square$

### A.2. Artificial case: Proof of Proposition A.2 and Proposition A.4

We recall the artificial case presented in Section 4 where we consider the distributions $\widetilde{P}$ and $\widetilde{Q}_\theta$ defined as follows.

We choose a target distribution $\widetilde{P} = \prod_{i=1}^{L} \widetilde{P}_{<i}$ where all factors $\widetilde{P}_{<i}$ are sparse uniform distributions over small subset of $K$ tokens, and a model distribution $\widetilde{Q}_\theta$ that matches $\widetilde{P}$ everywhere except at the two specific positions in the sequence $l_1$ and $l_2$, i.e, $\forall \boldsymbol{x} \in \mathcal{V}^L$, $\forall i \notin \{l_1, l_2\}$, $\widetilde{Q}_\theta(\cdot \mid \boldsymbol{x}_{<i}) = \widetilde{P}(\cdot \mid \boldsymbol{x}_{<i})$ . For position $l_1$ and $l_2$, the conditional distrbutions are defined as follows:

$$\widetilde{Q}_\theta(x \mid \boldsymbol{x}_{<l_1}) = \begin{cases} \frac{a}{\rho K}, & \text{if } x < \rho K, \\ \frac{b}{\rho K}, & \text{if } \rho K \le x \le K, \\ 0, & \text{otherwise}, \end{cases}$$

$$\widetilde{Q}_\theta(x \mid \boldsymbol{x}_{<l_2}) = \begin{cases} \frac{1-\epsilon}{K}, & \text{if } \widetilde{P}_{<l_2}(x) \ne 0, \\ \frac{\epsilon}{V-K}, & \text{otherwise}, \end{cases}$$

where $\rho \in [0,1]$ controls the number of tokens that will have extra-mass assigned under $\widetilde{Q}_\theta$, $0 \le a \le \lfloor \rho K \rfloor$ controls the excess mass assigned to the these tokens, and $b \ge 0$ is chosen to ensure normalization. $\epsilon \in [0, 1/2]$ determines the level of noise introduced outside the support of $\widetilde{P}_{<l_2}$.

We first present the proposition that fully characterizes the PR-Curve of $\widetilde{Q}_\theta$, along with its evolution under temperature scaling. We drop the $\sim$ notation for simplicity, and we denote $P = \widetilde{P}$ and $Q_\theta = \widetilde{Q}_\theta$.

**Proposition A.2** (PR-Curve under Temperature Scaling). *Let $P, Q_\theta \in \mathcal{P}(\mathcal{X}_V^K)$ respectively defined in the artifical case in Section 4. Then, for any temperature $t \in \mathbb{R}$, there exists a trade-off $\lambda_{\min}$ and $\lambda_{\max}$, with $\mu = \rho/(1-\rho)$:*

$$\lambda_{\min}^t = \frac{1}{1-\rho} \frac{(1-a)^{1/t}}{(1-a)^{1/t} + \mu^{1-1/t}a^{1/t}} \frac{(1-\epsilon)^{1/t}}{(1-\epsilon)^{1/t} + (V/K-1)^{1-1/t}\epsilon^{1/t}} \tag{33}$$
*and*
$$\lambda_{\max}^t = \frac{\mu^{-1/t}}{1-\rho} \frac{a^{1/t}}{(1-a)^{1/t} + \mu^{1-1/t}a^{1/t}} \frac{(1-\epsilon)^{1/t}}{(1-\epsilon)^{1/t} + (V/K-1)^{1-1/t}\epsilon^{1/t}}, \tag{34}$$

*such that the Precision $\alpha_\lambda(P\|Q_\theta^t)$ and Recall $\beta_\lambda(P\|Q_\theta^t)$ can be written as:*

* *For all $\lambda \ge \lambda_{\max}$:*

$$\alpha_\lambda(P\|Q_\theta^t) = \frac{(1-\epsilon)^{1/t}}{(1-\epsilon)^{1/t} + (V/K-1)^{1-1/t}\epsilon^{1/t}}. \tag{35}$$

$$\beta_\lambda(P\|Q_\theta^t) = \frac{1}{\lambda} \frac{(1-\epsilon)^{1/t}}{(1-\epsilon)^{1/t} + (V/K-1)^{1-1/t}\epsilon^{1/t}}. \tag{36}$$

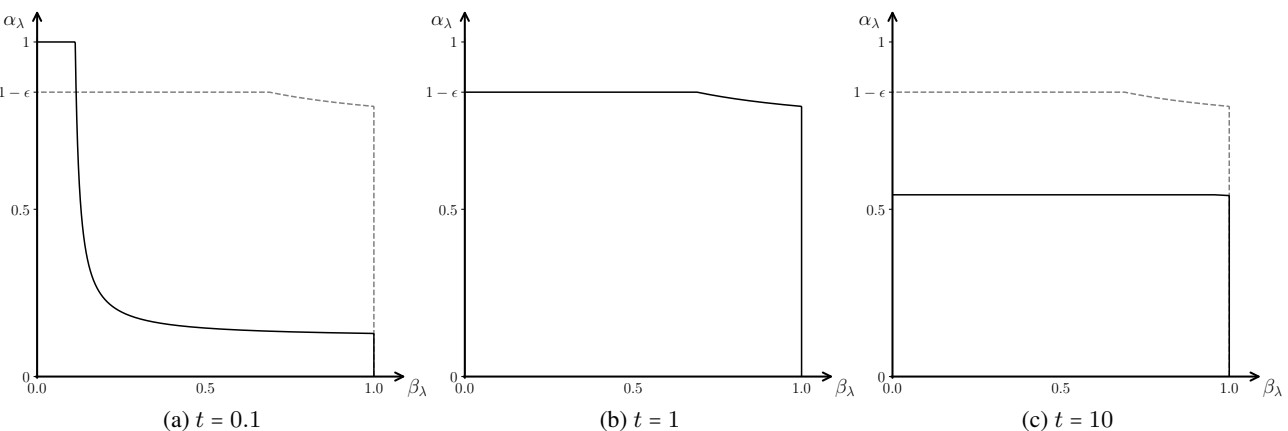

(a) $t = 0.1$       (b) $t = 1$       (c) $t = 10$

Figure 9: PR-Curve for different temperatures $t$ with $V = 100$, $K = 50$, $a/\rho = 1.45$, $\epsilon = 0.15$ and $\rho = 0.5$. The PR-Curve at $t = 1$ is represented in dashed line.

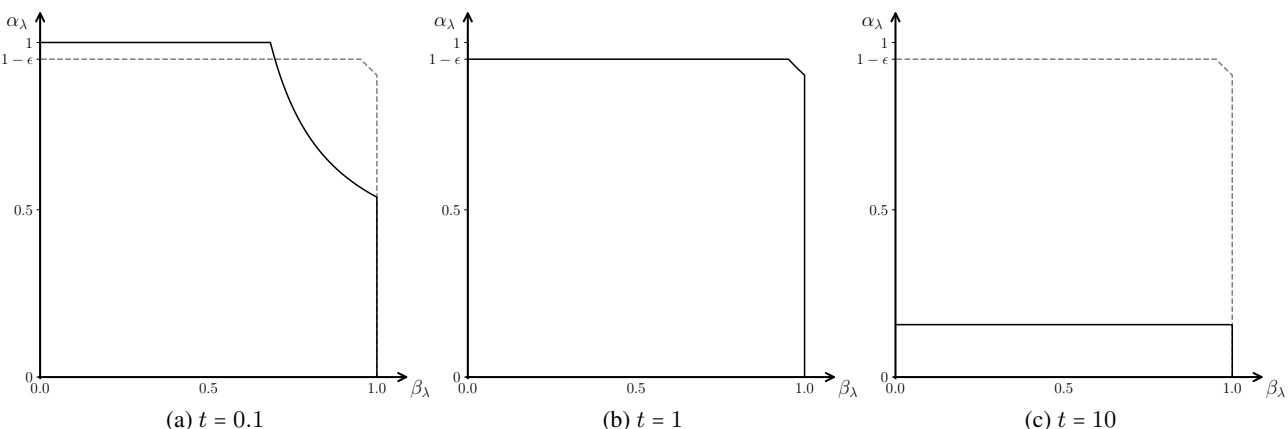

(a) $t = 0.1$       (b) $t = 1$       (c) $t = 10$

Figure 10: PR-Curve for different temperatures $t$ with $V = 100$, $K = 10$, $a/\rho = 1.05$, $\epsilon = 0.05$ and $\rho = 0.5$.

- *For all $\lambda_{\max} \geq \lambda \geq \lambda_{\min}$:*

$$\alpha_\lambda(P\|Q_\theta^t) = \rho\lambda + \frac{(1-a)^{1/t}}{(1-a)^{1/t} + \mu^{1-1/t}a^{1/t}} \frac{(1-\epsilon)^{1/t}}{(1-\epsilon)^{1/t} + (V/K-1)^{1-1/t}\epsilon^{1/t}}. \tag{37}$$

$$\beta_\lambda(P\|Q_\theta^t) = \rho + \frac{1}{\lambda} \frac{(1-a)^{1/t}}{(1-a)^{1/t} + \mu^{1-1/t}a^{1/t}} \frac{(1-\epsilon)^{1/t}}{(1-\epsilon)^{1/t} + (V/K-1)^{1-1/t}\epsilon^{1/t}}. \tag{38}$$

- *For all $\lambda \leq \lambda_{\min}$:*

$$\alpha_\lambda(P\|Q_\theta^t) = \lambda. \tag{39}$$

$$\beta_\lambda(P\|Q_\theta^t) = 1. \tag{40}$$

We can visualize the PR-Curve for different temperatures in Figures 9 and 10 with different parameters $V$, $K$, $a$, $\epsilon$ and $\rho$.

*Proof.* We begin by computing the PR-Curves for the original distribution $\widetilde{Q}_\theta$ at temperature $t = 1$. We then apply temperature scaling to obtain the tempered distributions and compute the corresponding PR curve.

**PR-Curve for $t = 1$.** Let's first compute Precision and Recall for extreme values of the trade-off parameter $\lambda$, i.e., $\lambda = +\infty$ and $\lambda = 0$.

- $\lambda = +\infty$:

$$\alpha_\infty(P\|Q_\theta) = Q_\theta(\mathrm{Supp}(P)) \tag{41}$$

$$= \sum_{\boldsymbol{x}\in\mathrm{Supp}(P)} Q_\theta(\boldsymbol{x}) \tag{42}$$

$$= \sum_{(x_1,\dots,x_L)\in\mathrm{Supp}(P)} \prod_l^L Q_\theta(x_l \mid \boldsymbol{x}_{<l}) \tag{43}$$

$$= \sum_{x_1\in\mathcal{S}(\boldsymbol{x}_{<l_1}),\dots,x_{l_1-1}\in\mathcal{S}(\boldsymbol{x}_{<l_1-1})} \prod_{l=1}^{l_1-1} Q_\theta(x_l \mid \boldsymbol{x}_{<l}) \times \sum_{x_{l_1}\in\mathcal{S}(\boldsymbol{x}_{<l_1})} Q_\theta(x_{l_1} \mid \boldsymbol{x}_{<l_1}) \tag{44}$$

$$\times \sum_{x_{l_1+1}\in\mathcal{S}(\boldsymbol{x}_{<l_1+1}),\dots,x_{l_2-1}\in\mathcal{S}(\boldsymbol{x}_{<l_2-1})} \prod_{l=l_1+1}^{l_2-1} Q_\theta(x_l \mid \boldsymbol{x}_{<l}) \times \sum_{x_{l_2}\in\mathcal{S}(\boldsymbol{x}_{<l_2})} Q_\theta(x_{l_2} \mid \boldsymbol{x}_{<l_2})$$

$$\times \sum_{x_{l_2+1}\in\mathcal{S}(\boldsymbol{x}_{<l_2+1}),\dots,x_L\in\mathcal{S}(\boldsymbol{x}_{<L})} \prod_{l=l_2+1}^{L} Q_\theta(x_l \mid \boldsymbol{x}_{<l})$$

$$= \sum_{x_1,\dots x_{l_1-1}} \prod_{l=1}^{l_1-1} P(x_l \mid \boldsymbol{x}_{<l}) \times \sum_{x_{l_1}\in\mathcal{S}(\boldsymbol{x}_{<l_1})} Q_\theta(x_{l_1} \mid \boldsymbol{x}_{<l_1}) \tag{45}$$

$$\times \sum_{x_{l_1+1}\in\mathcal{S}(\boldsymbol{x}_{<l_1+1}),\dots,x_{l_2-1}\in\mathcal{S}(\boldsymbol{x}_{<l_2-1})} \prod_{l=l_1+1}^{l_2-1} P(x_l \mid \boldsymbol{x}_{<l}) \times \sum_{x_{l_2}\in\mathcal{S}(\boldsymbol{x}_{<l_2})} Q_\theta(x_{l_2} \mid \boldsymbol{x}_{<l_2})$$

$$\times \sum_{x_{l_2+1}\in\mathcal{S}(\boldsymbol{x}_{<l_2+1}),\dots,x_L\in\mathcal{S}(\boldsymbol{x}_{<L})} \prod_{l=l_2+1}^{L} P(x_l \mid \boldsymbol{x}_{<l})$$

$$= \sum_{x_1,\dots x_{l_1-1}} \prod_{l=1}^{l_1-1} P(x_l \mid \boldsymbol{x}_{<l}) \times \sum_{l=1}^{K} \frac{c_l}{\rho K} \quad \text{where } c_l \text{ is either } a \text{ or } b \tag{46}$$

$$\times \sum_{x_{l_1+1}\in\mathcal{S}(\boldsymbol{x}_{<l_1+1}),\dots,x_{l_2-1}\in\mathcal{S}(\boldsymbol{x}_{<l_2-1})} \prod_{l=l_1+1}^{l_2-1} P(x_l \mid \boldsymbol{x}_{<l}) \times \sum_{x_{l_2}\in\mathcal{S}(\boldsymbol{x}_{<l_2})} \frac{1-\epsilon}{K}$$

$$\times \sum_{x_{l_2+1}\in\mathcal{S}(\boldsymbol{x}_{<l_2+1}),\dots,x_L\in\mathcal{S}(\boldsymbol{x}_{<L})} \prod_{l=l_2+1}^{L} P(x_l \mid \boldsymbol{x}_{<l})$$

$$= \sum_{l=1}^{K} \frac{c_l}{\rho K}(1-\epsilon) \tag{47}$$

$$= 1 - \epsilon. \tag{48}$$

Equation (45) is obtained under the assumption that $Q_\theta$ perfectly matches all conditional distributions of $P$ except $P(\cdot \mid \boldsymbol{x}_{<l_1})$ and $P(\cdot \mid \boldsymbol{x}_{<l_2})$. Equation (46) is derived by substituting the expression of $Q_\theta$. Equation (47) is obtained by iteratively marginalizing over the values of $x_1,\dots,x_{l_1-1}, x_{l_1+1},\dots,x_{l_2-1}$, and $x_{l_2+1},\dots,x_L$.

- $\lambda = 0$:

$$\beta_0(P\|Q_\theta) = P(\mathrm{Supp}(Q_\theta)) = 1 \tag{49}$$

since $\mathrm{Supp}(Q_\theta) \subset \mathrm{Supp}(P)$.

- $\lambda \in ]0, +\infty[$:

In the following, since $\alpha_\lambda = \lambda\beta_\lambda$, we focus our analysis on $\alpha_\lambda$.

$$\alpha_\lambda(P\|Q_\theta) = \sum_{\boldsymbol{x}\in\mathcal{X}_V^L} \min\left(\lambda P(\boldsymbol{x}),\ Q_\theta(\boldsymbol{x})\right) \tag{50}$$

$$= \sum_{x_1\ldots x_L} \min\left(\lambda \prod_{l=1}^L P(x_l\mid\boldsymbol{x}_{<l}),\ \prod_{l=1}^L Q_\theta(x_l\mid\boldsymbol{x}_{<l})\right) \tag{51}$$

$$= \sum_{\substack{x_1\ldots x_L}} \prod_{\substack{l=1\\l\neq l_1\\l\neq l_2}}^L P(x_l\mid\boldsymbol{x}_{<l}) \min\left(\lambda P(x_{l_1}\mid\boldsymbol{x}_{<l_1})P(x_{l_2}\mid\boldsymbol{x}_{<l_2}),\ Q_\theta(x_{l_1}\mid\boldsymbol{x}_{<l_1})Q_\theta(x_{l_2}\mid\boldsymbol{x}_{<l_2})\right) \tag{52}$$

$$= \sum_{\substack{x_1\ldots x_{l_2}}} \prod_{\substack{l=1\\l\neq l_1}}^{l_2-1} P(x_l\mid\boldsymbol{x}_{<l}) \tag{53}$$

$$\times\min\left(\lambda\left(\frac{1}{K}\mathbb{1}_{\{x_{l_1}\le K\}}\right)\left(\frac{1}{K}\mathbb{1}_{\{x_{l_2}\in\mathcal{S}(\boldsymbol{x}_{<l_2})\}}\right),\right.$$
$$\left.\left(\frac{c}{\rho K}\mathbb{1}_{\{x_{l_1}\le K\}}\right)\left(\mathbb{1}_{\{x_{l_2}\in\mathcal{S}(\boldsymbol{x}_{<l_2})\}}\frac{1-\epsilon}{K} + \mathbb{1}_{\{x_{l_2}\notin\mathcal{S}(\boldsymbol{x}_{<l_2})\}}\frac{\epsilon}{V-K}\right)\right),$$

where $c$ is either $a$ or $b$ depending on the value of $x_{l_1}$. When $x_{l_2}\notin\mathcal{S}(\boldsymbol{x}_{<l_2})$, the minimum is reached for $\lambda\frac{1}{V}\mathbb{1}_{\{x_2\in\mathcal{S}(\boldsymbol{x}_{<l_2})\}} = 0$. Therefore, the only terms that contribute to the sum are those for which $x_{l_2}\in\mathcal{S}(\boldsymbol{x}_{<l_2})$, i.e., the $K$ acceptable tokens at position $l_2$.

$$\alpha_\lambda(P\|Q_\theta) = \sum_{x_1\ldots x_{l_1}} \prod_{l=1}^{l_1-1} P(x_l\mid\boldsymbol{x}_{<l}) K\min\left(\frac{\lambda}{K^2}\mathbb{1}_{\{x_{l_1}\le K\}},\ \frac{c_l}{\rho K^2}\mathbb{1}_{\{x_{l_1}\le K\}}(1-\epsilon)\right) \tag{54}$$

$$= \sum_{x_1\ldots x_{l_1-1}} \prod_{l=1}^{l_1-1} P(x_l\mid\boldsymbol{x}_{<l}) \sum_l^K \min\left(\frac{\lambda}{K},\ \frac{c_l}{\rho K}(1-\epsilon)\right). \tag{55}$$

And finally by marginalizing over $x_1,\ldots,x_{l_1-1}$, we have:

$$\alpha_\lambda(P\|Q_\theta) = \frac{1}{K}\sum_i^K \min\left(\lambda,\ c_l(1-\epsilon)/\rho\right). \tag{56}$$

Three regimes can be distinguished:

1. For $\lambda \ge a(1-\epsilon)/\rho$, we have:

$$\alpha_\lambda(P\|Q_\theta) = \frac{1}{K}\sum_i^K c_l(1-\epsilon) = (1-\epsilon)\sum_i^K \frac{c_l}{\rho K} = 1-\epsilon. \tag{57}$$

In this regime, that is, for large values of $\lambda$, $\alpha_\lambda$ reflects the model's quality, as it converges to Precision when $\lambda\to+\infty$.

2. For $b(1-\epsilon)/\rho \le \lambda < a(1-\epsilon)/\rho$, we have:

$$\alpha_\lambda(P\|Q_\theta) = \frac{1}{K}\sum_{l=1}^{\rho K}\lambda + \frac{1}{K}\sum_{l=\rho K}^K b(1-\epsilon)/\rho \tag{58}$$

$$= \rho\lambda + b(1-\epsilon)\frac{1-\rho}{\rho}. \tag{59}$$

We will denote $\mu = \rho/(1-\rho)$ in the following, and since $\rho K\frac{a}{\rho K} + (1-\rho)K\frac{b}{\rho K} = 1$, we have:

$$b\frac{1-\rho}{\rho} = 1-a. \tag{60}$$

Thus, we have:

$$\alpha_\lambda(P\|Q_\theta) = \rho\lambda + (1-\epsilon)(1-a) \tag{61}$$

3. For $\lambda < b(1 - \epsilon)/\rho$, we have:

$$\alpha_\lambda(P\|Q_\theta) = \frac{1}{K}\sum_{l=1}^{K}\lambda = \lambda, \tag{62}$$

and therefore:

$$\beta_\lambda(P\|Q_\theta) = 1. \tag{63}$$

In that regime, Recall is maximal since the model generates all tokens.

**PR-Curve for Tempered distributions.**

- **Tempered distribution $Q_\theta^t$**: To simplify the notation, we define the inverse temperature $\tau = 1/t$ and set $\mu = \rho/(1 - \rho)$. We define the tempered distribution $Q_\theta^t$ as follows:

$$Q_\theta^t(x_{l_1} \mid \boldsymbol{x}_{<l_1}) = \frac{\left(\frac{c_l}{\rho K}\right)^\tau}{\sum_{j=1}^{K}\left(\frac{c_j}{K\rho}\right)^\tau + \sum_{j=K+1}^{V}(0)^\tau} \quad \text{where } c_l \text{ is } a \text{ for } l \leq \rho K \text{ and } b \text{ otherwise,} \tag{64}$$

$$= \frac{\left(\frac{1}{\rho K}\right)^\tau c_l^\tau}{\sum_{j=1}^{\rho K}\left(\frac{1}{\rho K}\right)^\tau a^\tau + \sum_{j=\rho K}^{K}\left(\frac{1}{\rho K}\right)^\tau b^\tau}, \tag{65}$$

$$= \frac{1}{K}\frac{c_l^\tau}{\rho a^\tau + (1 - \rho)b^\tau} \tag{66}$$

$$= \frac{1}{(1 - \rho)K}\frac{c_l^\tau}{\mu a^\tau + b^\tau}, \tag{67}$$

Moreover, since $\rho K \times \frac{a}{\rho K} + (1 - \rho)K \times \frac{b}{\rho K} = 1$, we have:

$$b = \frac{\rho}{1 - \rho} - \frac{\rho}{1 - \rho}a = \mu(1 - a). \tag{68}$$

Thus:

$$Q_\theta^t(x_{l_1} \mid \boldsymbol{x}_{<l_1}) = \frac{1}{(1 - \rho)K}\frac{c_l^\tau}{\mu a^\tau + \mu^\tau(1 - a)^\tau} \tag{69}$$

$$= \frac{1}{(1 - \rho)K}\frac{c_l^\tau}{\mu^\tau(1 - a)^\tau + \mu a^\tau} \tag{70}$$

$$= \frac{1}{(1 - \rho)K\mu^\tau}\frac{c_l^\tau}{(1 - a)^\tau + \mu^{1-\tau}a^\tau}. \tag{71}$$

Therefore, we have:

$$Q_\theta^t(x_{l_1} \mid \boldsymbol{x}_{<l_1}) = \begin{cases} \frac{1}{(1-\rho)K\mu^\tau}\frac{a^\tau}{(1-a)^\tau+\mu^{1-\tau}a^\tau} & \text{if } x_{l_1} \leq \rho K, \\ \frac{1}{(1-\rho)K}\frac{(1-a)^\tau}{(1-a)^\tau+\mu^{1-\tau}a^\tau} & \text{otherwise.} \end{cases} \tag{72}$$

For $x_{l_2} \in \mathcal{S}(\boldsymbol{x}_{<l_2})$:

$$Q_\theta^t(x_{l_2} \mid \boldsymbol{x}_{<l_2}) = \frac{\left(\frac{1-\epsilon}{K}\right)^\tau}{K\left(\frac{1-\epsilon}{K}\right)^\tau + (V - K)\left(\frac{\epsilon}{V-K}\right)^\tau} \tag{73}$$

$$= \frac{1}{K}\frac{\left(\frac{1}{K}\right)^\tau(1 - \epsilon)^\tau}{\left(\frac{1}{K}\right)^\tau(1 - \epsilon)^\tau + (V/K - 1)\left(\frac{\epsilon}{V-K}\right)^\tau} \tag{74}$$

$$= \frac{1}{K}\frac{(1 - \epsilon)^\tau}{(1 - \epsilon)^\tau + (V/K - 1)^{1-\tau}(\epsilon)^\tau}. \tag{75}$$

- **PR-Curve for $P$ and $Q_\theta^t$**: With these expressions, we can compute the PR-Curve for different values of $\tau$. By analogy with the previous computations at $t = 1$, we marginalize over all tokens for which $P^t(\cdot \mid \boldsymbol{x}_{<l}) = Q_\theta^t(\cdot \mid \boldsymbol{x}_{<l})$, yielding:

$$\alpha_\lambda(P\|Q_\theta^t) = \sum_{\boldsymbol{x}\in\mathcal{X}_V^L} \min\big(\lambda P(\boldsymbol{x}_l), Q_\theta^t(\boldsymbol{x})\big) \tag{76}$$

$$= \sum_{l=1}^{K} K \min\left(\frac{\lambda}{K^2}, \frac{1}{(1-\rho)K^2\mu^\tau}\frac{c_l^\tau}{(1-a)^\tau+\mu^{1-\tau}a^\tau}\frac{(1-\epsilon)^\tau}{(1-\epsilon)^\tau+(V/K-1)^{1-\tau}\epsilon^\tau}\right) \tag{77}$$

$$= \frac{1}{K}\sum_{l=1}^{K} \min\left(\lambda, \frac{1}{(1-\rho)\mu^\tau}\frac{c_l^\tau}{(1-a)^\tau+\mu^{1-\tau}a^\tau}\frac{(1-\epsilon)^\tau}{(1-\epsilon)^\tau+(V/K-1)^{1-\tau}\epsilon^\tau}\right). \tag{78}$$

Let us define:

$$\lambda_{\min}^t = \frac{1}{1-\rho}\frac{(1-a)^\tau}{(1-a)^\tau+\mu^{1-\tau}a^\tau}\frac{(1-\epsilon)^\tau}{(1-\epsilon)^\tau+(V/K-1)^{1-\tau}\epsilon^\tau} \tag{79}$$

and

$$\lambda_{\max}^t = \frac{\mu^{-\tau}}{1-\rho}\frac{a^\tau}{(1-a)^\tau+\mu^{1-\tau}a^\tau}\frac{(1-\epsilon)^\tau}{(1-\epsilon)^\tau+(V/K-1)^{1-\tau}\epsilon^\tau}. \tag{80}$$

We can show that there exists three regimes for the Precision and Recall dependent on the value of $\lambda$ compared to $\lambda_{\min}^t$ and $\lambda_{\max}^t$:

1. For $\lambda \geq \lambda_{\max}^t$, we have:

$$\min\left(\lambda, \frac{1}{(1-\rho)\mu^\tau}\frac{c_l^\tau}{(1-a)^\tau+\mu^{1-\tau}a^\tau}\frac{(1-\epsilon)^\tau}{(1-\epsilon)^\tau+(V/K-1)^{1-\tau}\epsilon^\tau}\right)$$
$$= \frac{1}{(1-\rho)\mu^\tau}\frac{c_l^\tau}{(1-a)^\tau+\mu^{1-\tau}a^\tau}\frac{(1-\epsilon)^\tau}{(1-\epsilon)^\tau+(V/K-1)^{1-\tau}\epsilon^\tau}.$$

Therefore:

$$\alpha_\lambda(P\|Q_\theta^t) = \frac{1}{K}\sum_{l}^{K}\frac{c_l^\tau}{\rho a^\tau+(1-\rho)b^\tau}\frac{(1-\epsilon)^\tau}{(1-\epsilon)^\tau+(V/K-1)^{1-\tau}\epsilon^\tau} \tag{81}$$

$$= \frac{1}{K}\frac{(1-\epsilon)^\tau}{(1-\epsilon)^\tau+(V/K-1)^{1-\tau}\epsilon^\tau}\sum_{l}^{K}\frac{c_l^\tau}{\rho a^\tau+(1-\rho)b^\tau} \tag{82}$$

$$= \frac{(1-\epsilon)^\tau}{(1-\epsilon)^\tau+(V/K-1)^{1-\tau}\epsilon^\tau}. \tag{83}$$

In particular:

$$\alpha_\infty(P\|Q_\theta^t) = \frac{(1-\epsilon)^\tau}{(1-\epsilon)^\tau+(V/K-1)^{1-\tau}\epsilon^\tau}.$$

We can observe that both:

$$\frac{\mu^{-\tau}}{1-\rho}\frac{a^\tau}{(1-a)^\tau+\mu^{1-\tau}a^\tau} = \frac{a^\tau}{\rho a^\tau+(1-\rho)b^\tau}$$

and

$$\frac{(1-\epsilon)^\tau}{(1-\epsilon)^\tau+(V-1)^{1-\tau}\epsilon^\tau},$$

are strictly decreasing functions of $t = 1/\tau$, since $a > b$ and $1 - \epsilon > \epsilon$. Therefore, in this regime, $\alpha_\lambda(P\|Q_\theta^t)$ is strictly decreasing with $t$. Moreover, the range of $\lambda$ values defining this regime is also strictly increasing, as $\lambda_{\max}^t$ decreases with $t$.

We can also observe that:

$$\lim_{t \to 0} \alpha_\lambda(P\|Q_\theta^t) = 1 \quad \text{and} \quad \lim_{t \to +\infty} \alpha_\lambda(P\|Q_\theta^t) = \frac{K}{V}. \tag{84}$$

Therefore, the range of $\lambda$ for which the Precision is constant is increases as the temperature increasing and the constant value of the $\alpha_\lambda$ is decreasing. Thus, the Precision is strictly decreasing from $1$ to $K/V$ as the temperature increases.

2. For $\lambda \in [\lambda_{\min}^t, \lambda_{\max}^t]$, we have:

$$\alpha_\lambda(P\|Q_\theta^t) = \sum_{l=1}^{\rho K} \frac{\lambda}{K} \tag{85}$$

$$+ \sum_{l=\rho K}^{K} \frac{1}{(1-\rho)K} \frac{(1-a)^\tau}{(1-a)^\tau + \mu^{1-\tau} a^\tau} \frac{(1-\epsilon)^\tau}{(1-\epsilon)^\tau + (V/K-1)^{1-\tau}\epsilon^\tau}$$

$$= \rho\lambda + \frac{(1-\rho)K}{(1-\rho)K} \frac{(1-a)^\tau}{(1-a)^\tau + \mu^{1-\tau} a^\tau} \frac{(1-\epsilon)^\tau}{(1-\epsilon)^\tau + (V/K-1)^{1-\tau}\epsilon^\tau} \tag{86}$$

$$= \rho\lambda + \frac{(1-a)^\tau}{(1-a)^\tau + \mu^{1-\tau} a^\tau} \frac{(1-\epsilon)^\tau}{(1-\epsilon)^\tau + (V/K-1)^{1-\tau}\epsilon^\tau}. \tag{87}$$

We can note that:

$$\alpha_\lambda(P\|Q_\theta^t) = \rho\lambda + (1-\rho)\lambda_{\min}^t. \tag{88}$$

In that regime, to know the behavior of the PR-Curve, we need to know if $\lambda_{\min}^t = \frac{(1-a)^\tau}{(1-a)^\tau + \mu^{1-\tau} a^\tau} \frac{(1-\epsilon)^\tau}{(1-\epsilon)^\tau + (V/K-1)^{1-\tau}\epsilon^\tau}$ is increasing or decreasing with $t$. However, we can observe that:

$$\lim_{t \to 0} \alpha(P\|Q_\theta^t) = \rho\lambda \quad \text{and} \quad \lim_{t \to +\infty} \alpha(P\|Q_\theta^t) = \rho\lambda + (1-\rho)K/V \tag{89}$$

3. For $\lambda \le \lambda_{\min}^t$, we have:

$$\alpha_\lambda(P\|Q_\theta^t) = \sum_{l=1}^{K} \frac{1}{K}\lambda = \lambda \text{ and thus } \beta_\lambda(P\|Q_\theta^t) = 1. \tag{90}$$

This concludes the proof.

$\square$

**Analysis of the PR-Curve.** To analyze the behavior of the PR-curve, we study the dependence of the expression

$$\frac{(1-a)^\tau}{(1-a)^\tau + \mu^{1-\tau} a^\tau} \cdot \frac{(1-\epsilon)^\tau}{(1-\epsilon)^\tau + (V/K-1)^{1-\tau}\epsilon^\tau}, \tag{91}$$

on the temperature parameter $t$.

We first introduce the function:

$$f_{\gamma,\nu}(\tau) := \frac{(1-\gamma)^\tau}{\nu^{1-\tau}\gamma^\tau + (1-\gamma)^\tau}, \tag{92}$$

which allows us to rewrite the above expression as $f_{a,\mu}(\tau) \cdot f_{\epsilon,V/K-1}(\tau)$.

Therefore, we focus on analyzing the behavior of the function $g(\tau) := f_{a,\mu}(\tau) \cdot f_{\epsilon,V/K-1}(\tau)$ as $\tau$ varies.

**Lemma A.3** (Rate for change of $g$)**.** *Let* $K, L \in \mathbb{N}^*$ *with* $K \le L$, $\epsilon \in ]0,1[$, $\rho \in ]0,1[$, $\mu = \rho/(1-\rho)$, $b \in [0,1]$ *and* $a = 1 - b/\mu$. *Then, we can define:*

$$\epsilon_0 := \frac{V/K - 1}{V/K - 1 + \left(\frac{a}{b}\right)^{\frac{\rho}{(1-K/V)}}}, \tag{93}$$

*such that:*

- *For all $\epsilon < \epsilon_0$, g is strictly decreasing with $\tau$.*

- *For all $\epsilon > \epsilon_0$, g is first decreasing then increasing with $\tau$.*

*Proof.* First, we can show that:

$$f'_{\gamma,\nu}(\tau) = \frac{\log(1-\gamma)(1-\gamma)^\tau \left[\nu^{1-\tau}\gamma^\tau + (1-\gamma)^\tau\right]}{(\nu^{1-\tau}\gamma^\tau + (1-\gamma)^\tau)^2} \tag{94}$$

$$- \frac{(1-\gamma)^\tau \left[\log(\gamma)\nu^{1-\tau}\gamma^\tau - \log(\nu)\nu^{1-\tau}\gamma^\tau + \log(1-\gamma)(1-\gamma)^\tau\right]}{(\nu^{1-\tau}\gamma^\tau + (1-\gamma)^\tau)^2}$$

$$= \frac{\nu^{1-\tau}\gamma^\tau(1-\gamma)^\tau \log\left(\frac{1-\gamma}{\gamma/\nu}\right)}{(\nu^{1-\tau}\gamma^\tau + (1-\gamma)^\tau)^2}. \tag{95}$$

$$\tag{96}$$

By noting that:

$$f_{1-\gamma,1/\nu}(\tau) = \frac{\gamma^\tau}{\gamma^\tau + (1/\nu)^{1-\tau}(1-\gamma)^\tau} \tag{97}$$

$$= \frac{\nu^{1-\tau}}{\nu^{1-\tau}} \frac{\gamma^\tau}{\gamma^\tau + (1/\nu)^{1-\tau}(1-\gamma)^\tau} \tag{98}$$

$$= \frac{\nu^{1-\tau}\gamma^\tau}{\nu^{1-\tau}\gamma^\tau + (1-\gamma)^\tau}, \tag{99}$$

we can write the derivative of $g$ as:

$$f'_{\gamma,\nu}(\tau) = \log\left(\frac{1-\gamma}{\gamma/\nu}\right) f_{\gamma,\nu}(\tau) f_{1-\gamma,1/\nu}(\tau). \tag{100}$$

Then:

$$g'(\tau) = f'_{a,\mu}(\tau) f_{\epsilon,V/K-1}(\tau) + f_{a,\mu}(\tau) f'_{\epsilon,V-1}(\tau) \tag{101}$$

$$= \log\left(\frac{1-a}{a/\mu}\right) f_{a,\mu}(\tau) f_{1-a,1/\mu}(\tau) f_{\epsilon,V/K-1}(\tau) \tag{102}$$

$$+ f_{a,\mu}(\tau) \log\left(\frac{1-\epsilon}{\epsilon/(V/K-1)}\right) f_{\epsilon,V/K-1}(\tau) f_{1-\epsilon,1/(V/K-1)}(\tau)$$

$$= f_{a,\mu}(\tau) f_{\epsilon,V/K-1}(\tau) \Bigg[ f_{1-a,1/\mu}(\tau) \log\left(\frac{1-a}{a/\mu}\right) \tag{103}$$

$$+ f_{1-\epsilon,1/(V/K-1)}(\tau) \log\left(\frac{1-\epsilon}{\epsilon/(V/K-1)}\right) \Bigg].$$

By observing that $1/f_{1-\gamma,1/\nu}(\tau) = (\nu^{1-\tau}\gamma^\tau + (1-\gamma)^\tau)/\nu^{1-\tau}\gamma^\tau = 1 + \nu^{\tau-1}\left(\frac{1-\gamma}{\gamma}\right)^\tau$, we can show that $g'(\tau)$ can be rewritten as:

$$g'(\tau) = f_{a,\mu}(\tau) f_{1-a,1/\mu}(\tau) f_{\epsilon,V/K-1}(\tau) f_{1-\epsilon,1/(V/K-1)}(\tau) \times \tag{104}$$

$$\left[ \left[ 1 + \mu^{\tau-1}\left(\frac{\mu(1-a)}{a}\right)^\tau \right] \log\left(\frac{1-\epsilon}{\epsilon/(V/K-1)}\right) \right.$$

$$\left. - \left[ 1 + (V/K-1)^{\tau-1}\left(\frac{1-\epsilon}{\epsilon}\right)^\tau \right] \log\left(\frac{a}{\mu(1-a)}\right) \right]$$

$$= f_{a,\mu}(\tau) f_{1-a,1/\mu}(\tau) f_{\epsilon,V/K-1}(\tau) f_{1-\epsilon,1/(V/K-1)}(\tau) \times h(1/\tau). \tag{105}$$

As $f_{a,\mu}(\tau)f_{1-a,1/\mu}(\tau)f_{\epsilon,V/K-1}(\tau)f_{1-\epsilon,1/(V/K-1)}(\tau) > 0$, we need to study the sign of $h(1/\tau) = h(t)$ with $t$. We can show that:

$$h(t) = \left[1 + \frac{1}{\mu}\left(\frac{1-a}{a/\mu}\right)^{1/t}\right]\log\left(\frac{1-\epsilon}{\epsilon/(V/K-1)}\right) \tag{106}$$

$$-\left[1 + \frac{1}{V/K-1}\left(\frac{1-\epsilon}{\epsilon/(V/K-1)}\right)^{1/t}\right]\log\left(\frac{a}{\mu(1-a)}\right),$$

where $\mu(1-a)/a = b/a < 1$ and $(1-\epsilon)/\epsilon(V/K-1) > 1/(V/K-1) > 1$. We will show that $h$ is strictly increasing. If we denote $r_{\gamma,\nu}(t) = ((1-\gamma)/(\gamma/\nu))^{1/t}$, we have:

$$r'_{\gamma,\nu}(t) = -\frac{1}{t^2}\log\left(\frac{1-\gamma}{\gamma/\nu}\right)r_{\gamma,\nu}(t). \tag{107}$$

Therefore, we can compute the derivative of $h$:

$$h'(t) = -\frac{1}{t^2}\log\left(\frac{\mu(1-a)}{a}\right)r_{a,\mu}(t)\log\left(\frac{1-\epsilon}{\epsilon/(V/-1)}\right) \tag{108}$$

$$+ \frac{1}{(V/K-1)t^2}\log\left(\frac{1-\epsilon}{\epsilon/(V/K-1)}\right)r_{\epsilon,V/K-1}(t)\log\left(\frac{a}{\mu(1-a)}\right)$$

$$= \frac{1}{t^2}\underbrace{\log\left(\frac{1-\epsilon}{\epsilon/(V/K-1)}\right)}_{>0}\underbrace{\log\left(\frac{a}{\mu(1-a)}\right)}_{>0}\underbrace{\left[\frac{r_{\epsilon,V/K-1}(t)}{V-1} + r_{a,\mu}(t)\right]}_{>0}. \tag{109}$$

Therefore, $h$ is strictly increasing. Considering the limits of $h$ when $t$ goes to $0$ and $+\infty$, we can show that:

$$\lim_{t\to 0} h(t) = -\infty \quad \text{since} \quad \begin{cases} \lim_{t\to 0}\left(\frac{\mu(1-a)}{a}\right)^{1/t} = 0, \\ \lim_{t\to 0}\left(\frac{1-\epsilon}{\epsilon/(V/K-1)}\right)^{1/t} = +\infty \end{cases} \tag{110}$$

and

$$\lim_{t\to+\infty} h(t) = \frac{1+\mu}{\mu}\log\left(\frac{1-\epsilon}{\epsilon/(V/K-1)}\right) + \frac{V}{V-K}\log\left(\frac{\mu(1-a)}{a}\right) \tag{111}$$

$$\text{since} \quad \begin{cases} \lim_{t\to+\infty}\left(\frac{\mu(1-a)}{a}\right)^{1/t} = 1, \\ \lim_{t\to+\infty}\left(\frac{1-\epsilon}{\epsilon(V-1)}\right)^{1/t} = 1. \end{cases}$$

To study the variation rate of $g(\tau)$, we need to study the sign of $\lim_{t\to+\infty} h(t)$. We can show that:

$$\lim_{t\to+\infty} h(t) \geq 0 \Leftrightarrow \frac{1+\mu}{\mu}\log(V/K-1) + \frac{1+\mu}{\mu}\log\left(\frac{1-\epsilon}{\epsilon}\right) + \frac{V}{V-K}\log\left(\frac{b}{a}\right) \geq 0 \tag{112}$$

$$\Leftrightarrow \frac{1+\mu}{\mu}\log(1/\epsilon-1) \geq \frac{V}{V-K}\log\left(\frac{a}{b}\right) - \frac{1+\mu}{\mu}\log(V/K-1) \tag{113}$$

$$\Leftrightarrow 1/\epsilon - 1 \geq \left(\frac{a}{b}\right)^{\frac{\mu V}{(1+\mu)(V-K)}}\left(\frac{1}{V/K-1}\right) \tag{114}$$

$$\Leftrightarrow \epsilon \leq \frac{V/K-1}{V/K-1 + \left(\frac{a}{b}\right)^{\frac{\rho V}{(V-K)}}}, \tag{115}$$

by noting that $\mu/(1+\mu) = \rho$. Therefore, by defining

$$\epsilon_0 = \frac{V/K-1}{V/K-1 + \left(\frac{a}{b}\right)^{\frac{\rho}{(1-K/V)}}}, \tag{116}$$

We can identify two cases for the rate of variation of $g$:

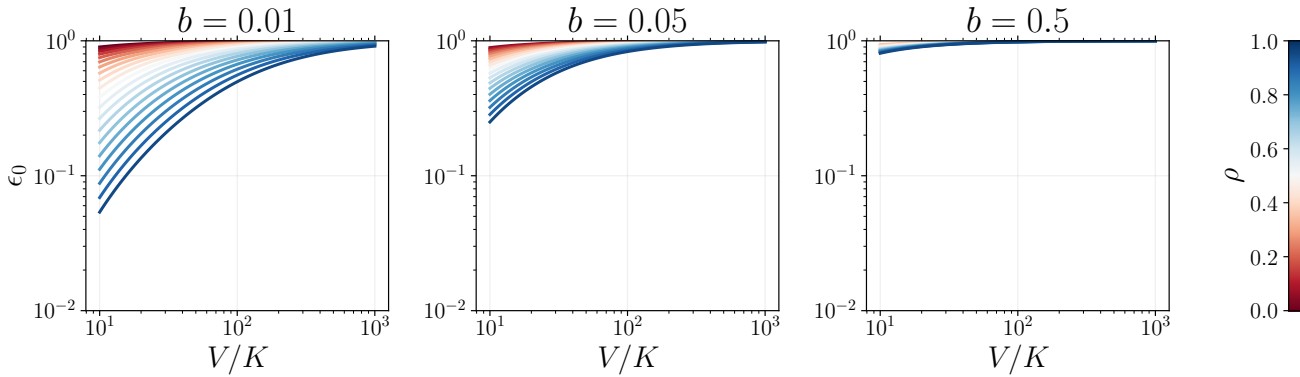

Figure 11: Values of $\epsilon_0$ for different values of $V/K$, $b$, and $\rho$. We consider different levels of token underrepresentation by varying $b$ and $\rho$. Low values of $b$ indicate that the model assigns very low probability to the relevant tokens, while low values of $\rho$ mean that a large proportion of tokens are underrepresented. We observe that even for small values of $V/K$, when $b$ is low and $\rho$ is high, the value of $\epsilon_0$ remains relatively large, often above 0.1. We recall that when $\epsilon < \epsilon_0$, Recall starts to decrease at some point as temperature increases.

- First when $\epsilon \geq \epsilon_0$, then $h(t) \leq 0$ for all $t$ and therefore $g$ is increasing with $t$.

- On the other side, if $\epsilon < \epsilon_0$, then there exists $t_0$ such that $h(t_0) = 0$ and thus such that $h(t) > 0$ for $t > t_0$ and $h(t) < 0$ for $t < t_0$. Therefore, $g$ is first increasing then decreasing.

$\square$

We can now analyze the behavior of the PR-Curve based on Lemma A.3:

**Proposition A.4** (Variation of PR-Curves with Temperature). *Let $P, Q_\theta \in \mathcal{P}(\mathcal{X}_V^K)$ be the distributions defined in the artificial setting of Section 4. Then, as the temperature $t$ increases:*

- *The threshold $\lambda_{\max}^t$ decreases strictly with $t$, satisfying:*

$$\lim_{t \to 0} \lambda_{\max}^t = +\infty \quad and \quad \lim_{t \to +\infty} \lambda_{\max}^t = \frac{K}{V}. \tag{117}$$

- *For the lower threshold $\lambda_{\min}^t$:*
    - *If $\epsilon > \epsilon_0$, then $\lambda_{\min}^t$ increases strictly with $t$.*
    - *If $\epsilon \leq \epsilon_0$, then $\lambda_{\min}^t$ increases for small $t$, then decreases after a certain point.*

    *In both cases:*

$$\lim_{t \to 0} \lambda_{\min}^t = 0 \quad and \quad \lim_{t \to +\infty} \lambda_{\min}^t = \frac{K}{V}. \tag{118}$$

*The behavior of the PR-Curve with respect to temperature depends on the regime of $\lambda$ in relation to $\lambda_{\min}^t$ and $\lambda_{\max}^t$:*

- ***High-$\lambda$ regime*** *($\lambda \geq \lambda_{\max}^t$): The interval $\{\lambda \geq \lambda_{\max}^t\}$ widens as $t$ grows. In this case, both Precision $\alpha_\lambda(P\|Q_\theta^t)$ and Recall $\beta_\lambda(P\|Q_\theta^t)$ decrease strictly with $t$, converging respectively to $K/V$ and $K/(V\lambda)$. More precisely, for any fixed $\lambda$, there exists a temperature $t_0$ such that for all $t \geq t_0$, Precision and Recall decrease strictly with temperature. The larger the value of $\lambda$, the smaller the corresponding $t_0$.*

- ***Low-$\lambda$ regime*** *($\lambda \leq \lambda_{\min}^t$):*
    - *If $\epsilon > \epsilon_0$, this regime expands with increasing $t$.*
    - *If $\epsilon \leq \epsilon_0$, the regime first expands, then contracts.*

*In this regime, both Precision and Recall are constant with respect to temperature, taking values $\alpha_\lambda = \lambda$ and $\beta_\lambda = 1$.*

- ***Intermediate-$\lambda$ regime** ($\lambda \in [\lambda_{\min}^t, \lambda_{\max}^t]$): As $t \to \infty$, this range collapses to $\lambda = K/V$. In this regime:*

    - *If $\epsilon > \epsilon_0$, both Precision and Recall increase strictly with temperature.*
    - *If $\epsilon \leq \epsilon_0$, both Precision and Recall increase initially with t, then decrease.*

    *In all cases, they converge to $\alpha_\lambda \to K/V$, $\beta_\lambda \to 1$.*

*Moreover, if $\frac{K}{V} \leq (1 - a)(1 - \epsilon)$, then there exists a temperature $t_0$ such that for all $t \geq t_0$:*

$$\beta_\lambda(P \| Q_\theta^t) < \beta_\lambda(P \| Q_\theta^1). \tag{119}$$

*Proof.* The different regimes follow directly from Lemma A.3 and the characterizations in Theorem A.2:

- The behavior in the high-$\lambda$ regime is derived from the asymptotic expressions in Eq. 84.

- For the intermediate regime, the PR-Curve values follow Eq. 88, and Lemma A.3 determines the qualitative behavior based on $\epsilon_0$.

- For the regime of low $\lambda$, only the Recall and Precision are fixed. Only the range of $\lambda$ defined by $\lambda < \lambda_{\min}^t$ varies and follows the same variations observed in Lemma A.3.

$\square$

# B. Proof of results in Section 5

## B.1. Proofs for Subsection 5.2: *Trunc*

**Trunc loss.** Define $P^{\text{Trunc}}(\cdot) = P(\cdot \mid \mathcal{X}_{\text{Trunc}})$, and let $H(\cdot)$ denote the entropy. Informally, the following proposition states that if $\mathcal{L}^{\Delta}_{\text{Trunc}}$ gets arbitrarily close to its minimum value $(1 - \Delta)H(P^{\text{Trunc}})$ then for a recall arbitrarily close to $1 - \Delta$, the precision will be arbitrarily close to $1$.

**Proposition.** *(Trunc optimizes the precision at a given recall) For any $\epsilon > 0$, if $\mathcal{L}^{\Delta}_{Trunc}(\theta) \leq \epsilon + (1 - \Delta)H(P^{Trunc})$, then there is a precision-recall pair $(\alpha, \beta)$ such that $\beta \geq (1 - \Delta) - \sqrt{\epsilon}$, and $\alpha \geq 1 - \sqrt{\frac{\epsilon}{2(1-\Delta)}}$.*

*Proof.* Let $\mathcal{X}_{\text{Trunc}} = \{x : Q(x) \geq \delta\}$ and $\Delta = 1 - P(\mathcal{X}_{\text{Trunc}})$. Let us first rewrite the objective function:

$$
\begin{aligned}
\mathcal{L}^{\Delta}_{\text{Trunc}}(\theta) &= -\mathbb{E}_{\boldsymbol{x} \sim P}\left[ \sum_{l=1}^{L} \mathbb{1}_{\{Q(\boldsymbol{x}) \geq \delta\}} \log Q_{<l}(x_l) \right] \\
&= -\mathbb{E}_{\boldsymbol{x} \sim P}\left[ \mathbb{1}_{\{Q(\boldsymbol{x}) \geq \delta\}} \log Q(\boldsymbol{x}) \right] \\
&= -P(\mathcal{X}_{\text{Trunc}}) \times \mathbb{E}_{\boldsymbol{x} \sim P(\cdot | \mathcal{X}_{\text{Trunc}})}\left[ \log Q(\boldsymbol{x}) \right] \\
&= -(1 - \Delta)\mathbb{E}_{\boldsymbol{x} \sim P^{\text{Trunc}}(\cdot)}\left[ \log Q(\boldsymbol{x}) \right] \\
&= (1 - \Delta)\mathcal{D}_{\text{KL}}\left( P^{\text{Trunc}} \| Q \right) + (1 - \Delta)H(P^{\text{Trunc}})
\end{aligned}
$$

Combining the inequality $\mathcal{L}^{\Delta}_{\text{Trunc}}(\theta) \leq \epsilon + (1 - \Delta)H(P^{\text{Trunc}})$ with Pinsker inequality, we have

$$
\mathcal{D}_{\text{TV}}\left( P^{\text{Trunc}}, Q \right) \leq \sqrt{\frac{\mathcal{D}_{\text{KL}}\left( P^{\text{Trunc}} \| Q \right)}{2}} \leq \sqrt{\frac{\epsilon}{2(1 - \Delta)}}
$$

Choosing $\lambda = \frac{1}{1-\Delta}$, we are now ready to compute lower bounds on Precision-Recall:

$$
\begin{aligned}
\alpha_\lambda(P \| Q) &= \sum_{\boldsymbol{x}} \min\left( \frac{1}{1 - \Delta}P(\boldsymbol{x}), Q(\boldsymbol{x}) \right) \\
&\geq \sum_{\boldsymbol{x} \in \mathcal{X}_{\text{Trunc}}} \min\left( \frac{1}{1 - \Delta}P(\boldsymbol{x}), Q(\boldsymbol{x}) \right) \\
&= \sum_{\boldsymbol{x} \in \mathcal{X}} \min\left( P^{\text{Trunc}}(\boldsymbol{x}), Q(\boldsymbol{x}) \right) = 1 - \mathcal{D}_{\text{TV}}(P^{\text{Trunc}}, Q) \\
&\geq 1 - \sqrt{\frac{\epsilon}{2(1 - \Delta)}} \\
\beta_\lambda(P \| Q) &= \frac{1}{\lambda}\alpha_\lambda(P \| Q) \geq (1 - \Delta) - \sqrt{\frac{\epsilon(1 - \Delta)}{2}}
\end{aligned}
$$

$\square$

**TruncR loss.** We proove Proposition 5.3 using characterization of the optimal distribution under the $\mathcal{L}^{1-\Delta}_{\text{TruncR}}$ loss.

We first characterize the optimal distribution minimizing the $\mathcal{L}^{1-\Delta}_{\text{TruncR}}$ loss. This result will serve as a foundational lemma for proving the main proposition relating TruncR optimization to Recall at fixed Precision.

**Lemma B.1** (Optimal TruncR distribution). *Let $P = (P_1, \ldots, P_n)$ be a discrete reference distribution. Consider the loss*

$$
\mathcal{L}^{1-\Delta}_{\text{TruncR}}(Q) = -\sum_{i=1}^{n} P_i \, \mathbb{1}_{\{Q_i < \delta\}} \log Q_i,
$$

*where $\delta > 0$ is chosen so that $\sum_{i:Q_i < \delta} P_i = 1 - \Delta$, and $Q = (Q_1, \ldots, Q_n)$ ranges over the simplex. Then the unique minimizer is*

$$
Q_i^* = \min\left( \delta, \, \gamma P_i \right),
$$

*where $\gamma > 0$ and $\delta$ satisfy*

$$\sum_{i=1}^{n} Q_i^* = 1, \qquad \sum_{i:\gamma P_i < \delta} P_i = 1 - \Delta.$$

*Proof.* Introduce Lagrange multiplier $\lambda > 0$ for the constraint $\sum_i Q_i = 1$. The Lagrangian is

$$\mathcal{J}(Q, \lambda) = -\sum_i P_i \mathbf{1}_{\{Q_i < \delta\}} \log Q_i + \lambda \Big(\sum_i Q_i - 1\Big).$$

Stationarity with respect to $Q_i$ gives

$$-\frac{P_i}{Q_i} \mathbf{1}_{\{Q_i < \delta\}} + \lambda \begin{cases} = 0 \implies Q_i = \frac{P_i}{\lambda}, \\ < 0 \implies Q_i = \delta. \end{cases}$$

Hence $Q_i^* = \min(\delta, P_i/\lambda)$. Setting $\gamma = 1/\lambda$ yields $Q_i^* = \min(\delta, \gamma P_i)$. Finally, choose $\gamma$ and $\delta$ so that $\sum_i Q_i^* = 1$ and $\sum_{i:\gamma P_i < \delta} P_i = 1 - \Delta$. Uniqueness follows from the strict concavity of $\sum_i P_i \log Q_i$. $\qquad\square$

*Remark* B.2 (Lower bound on $\gamma$). Let

$$S = \{i : \gamma P_i < \delta\}, \quad S^c = \{1, \dots, n\} \smallsetminus S.$$

The quantile constraint gives $\sum_{i \in S} P_i = 1 - \Delta$.

Normalization gives

$$1 = \sum_{i \in S} \gamma P_i + \sum_{i \in S^c} \delta = \gamma(1 - \Delta) + |S^c| \delta.$$

At the boundary of $S$, $\delta = \gamma P_{\max(S)}$. Thus

$$1 = \gamma \big[(1 - \Delta) + P_{\max(S)} |S^c|\big] \implies \gamma = \frac{1}{(1 - \Delta) + P_{\max(S)} |S^c|} \geq 1,$$

since $(1 - \Delta) + P_{\max(S)} |S^c| \leq (1 - \Delta) + \sum_{i \in S^c} P_i = 1$.

We now state and prove the main proposition linking the minimization of the $\mathcal{L}_{\mathrm{TruncR}}^{1-\delta}$ loss to Recall maximization under a fixed Precision constraint.

**Proposition B.3** (TruncR minimization favors Recall at fixed Precision). *Let $P = (P_1, \dots, P_n)$ be a reference distribution and $Q^* = \min(\delta, \gamma P)$ be the optimal distribution minimizing the TruncR loss $\mathcal{L}_{\mathrm{TruncR}}^{1-\Delta}$, from Lemma B.1. Then, for all $\Delta > 0$, any $\lambda > 0$ such that $\alpha_\lambda(P \| Q^*) = 1 - \Delta$ satisfies*

$$\beta_\lambda(P \| Q^*) \geq 1 - \Delta.$$

*In particular, minimizing the TruncR loss with constraint $\sum_{i:Q_i < \delta} P_i = 1 - \Delta$ leads to a distribution achieving Recall at least $1 - \Delta$ at Precision level $1 - \Delta$.*

*Proof.* We compute the Precision for $Q^*$. For any $\lambda > 0$, define $\eta := \min(\lambda, \gamma)$. Then

$$\alpha_\lambda(P \| Q^*) = \sum_i \min(\lambda P_i, Q_i^*) = \sum_i \min(\lambda P_i, \min(\gamma P_i, \delta)) \tag{120}$$

$$= \sum_i \min(\eta P_i, \delta). \tag{121}$$

First, observe that if $\lambda > \gamma$, then $\eta = \gamma$ and $\alpha_\lambda = \sum_i Q_i^* = 1$, by the normalization of $Q^*$. So for any $\Delta > 0$, a solution with $\alpha_\lambda = 1 - \Delta < 1$ must necessarily satisfy $\lambda \leq \gamma$.

Since $\gamma \geq 1$ by Remark B.2, we are in the regime where $\lambda \leq 1 \leq \gamma$, and thus $\eta = \lambda$. In particular, when $\lambda = 1$,

$$\beta_1 = \sum_i \min(P_i, \delta) \tag{122}$$

$$= \sum_{i \in S} \min(P_i, \delta) + \sum_{i \notin S} \min(P_i, \delta) \tag{123}$$

$$= \sum_{i \in S} P_i + \sum_{i \notin S} \min(P_i, \delta), \tag{124}$$

$$= 1 - \Delta + \sum_{i \notin S} \min(P_i, \delta) \tag{125}$$

$$\geq 1 - \Delta, \tag{126}$$

where in Eq. (124) we used the fact that for $i \in S = \{i : \gamma P_i < \delta\}$, we have $P_i < \delta$, so $\min(P_i, \delta) = P_i$.

Now, let $\lambda$ be such that $\alpha_\lambda = 1 - \Delta$. Then from the above, $\lambda \leq 1$, and thus

$$\beta_\lambda(P\|Q^*) = \frac{1}{\lambda}\alpha_\lambda = \frac{1-\Delta}{\lambda} \geq 1 - \Delta.$$

This shows that the Recall corresponding to Precision level $1 - \Delta$ is necessarily at least $1 - \Delta$, which concludes the proof. $\quad\square$

## B.2. Proofs for Subsection 5.3: *GOLD*

Let us recall the Theorem 5.4:

**Theorem** ($\alpha$-Divergence Minimization). *Minimizing the $\alpha$-divergences between $P_{<l}$ and $Q_{<l}$ for all $l \in \{1, \ldots, L\}$ is equivalent to minimizing*

$$\mathcal{L}_{c\text{-}Div}^\alpha(\theta) = -\mathbb{E}_{\boldsymbol{x} \sim P}\left[\sum_{l=1}^L \bar{Q}_\theta(x_l|\boldsymbol{x}_{<l})^{1-\alpha} \log Q_\theta(x_l|\boldsymbol{x}_{<l})\right]. \tag{127}$$

*Proof.* Let us focus on the $\alpha$-divergence $\mathcal{D}_\alpha$ any distributions $P_{<l}$ and $Q_{<l}$ and $x_l \sim P_{<l}$:

$$\mathcal{D}_\alpha(P_{<l}\|Q_{<l}) = \frac{1}{\alpha-1}\left[\sum_{x \in \mathcal{V}} P_{<l}(x)^\alpha Q_{<l}(x)^{1-\alpha} - 1\right] \tag{128}$$

$$= \frac{1}{\alpha-1}\left[Q_{<l}(x_l)^{1-\alpha} - 1\right] \tag{129}$$

by assuming that $P_{<l} = \widehat{P}_{<l}$ with $\gamma = 1$. We can write the gradient of the $\alpha$-divergence with respect to the model parameters $\theta$:

$$\nabla_\theta \mathcal{D}_\alpha(P\|Q_\theta) = \frac{1}{\alpha-1}\nabla_\theta\left[Q_\theta(x_l)^{1-\alpha} - 1\right] \tag{130}$$

$$= \frac{1-\alpha}{\alpha-1}Q_\theta(x_l)^{-\alpha}\nabla_\theta Q_\theta(x_l) \quad \text{by the chain rule,} \tag{131}$$

$$= -Q_\theta(x_l)^{-\alpha+1}\frac{1}{Q_\theta(x_l)}\nabla_\theta Q_\theta(x_l) \tag{132}$$

$$= -Q_\theta(x_l)^{-\alpha+1}\nabla_\theta \log Q_\theta(x_l) \quad \text{since } \nabla_\theta \log Q_\theta(x_l) = \nabla_\theta Q_\theta(x_l)/Q_\theta(x_l) \tag{133}$$

$$= Q_\theta(x_l)^{-\alpha+1}\nabla_\theta\left[-\log Q_\theta(x_l)\right]. \tag{134}$$

Therefore, minimizing the $\alpha$-divergence is equivalent to maximizing the loss:

$$\bar{Q}(x_l)^{1-\alpha} \times \left[-\log Q(x_l)\right]. \tag{135}$$

$$\square$$

## B.3. Proofs for Subsection 5.4: *TaiLr*

First let us recall proposition 5.5:

**Proposition.** *The optimal distribution $Q_\theta$ using the TaiLr method with $\gamma > 0$ satisfies the following property:*

$$\forall \boldsymbol{x} \in \mathcal{V}^L, \ \forall l \in 1, \ldots, L, \quad Q_\theta(x_l|\boldsymbol{x}_{<l}) = \frac{P(x_l|\boldsymbol{x}_{<l})(1 - \gamma + V\gamma) - \gamma}{1 - \gamma}. \tag{136}$$

*In others words, $Q(x_l|\boldsymbol{x}_{<l}) > P(x_l|\boldsymbol{x}_{<l})$ if $P(x_l|\boldsymbol{x}_{<l}) > 1/V$ and $Q(x_l|\boldsymbol{x}_{<l}) \le P(x_l|\boldsymbol{x}_{<l})$ otherwise.*

*Proof.* The loss minimized by the *TaiLr* method is:

$$\mathcal{L}_{\text{TaiLr}}^\gamma(\theta) = -\mathbb{E}_{\boldsymbol{x} \sim P}\left[\sum_{l=1}^L \frac{\bar{Q}_{<l}(x_l)}{\gamma + (1-\gamma)\bar{Q}_{<l}(x_l)} \log Q_\theta(x_l|\boldsymbol{x}_{<l})\right]. \tag{137}$$

Differentiating every term with respect to $\theta$, we have:

$$\nabla_\theta\left[\frac{\bar{Q}_{<l}(x_l)}{\gamma + (1-\gamma)\bar{Q}_{<l}(x_l)} \log Q_\theta(x_l|\boldsymbol{x}_{<l})\right] = \frac{\bar{Q}_{<l}(x_l)\nabla_\theta \log Q_\theta(x_l|\boldsymbol{x}_{<l})}{\gamma + (1-\gamma)\bar{Q}_{<l}(x_l)} \tag{138}$$

$$= \frac{1-\gamma}{1-\gamma}\frac{\nabla_\theta Q_\theta(x_l|\boldsymbol{x}_{<l})}{\gamma + (1-\gamma)\bar{Q}_{<l}(x_l)} \tag{139}$$

$$= \frac{1}{1-\gamma}\nabla_\theta \log(\gamma + (1-\gamma)Q_\theta(x_l|\boldsymbol{x}_{<l})). \tag{140}$$

Minimizing $\mathcal{L}_{\text{TaiLr}}^\gamma(\theta)$ is equivalent to maximizing the following loss:

$$\mathbb{E}_{\boldsymbol{x} \sim P}\left[\sum_{l=1}^L \log(\gamma + (1-\gamma)Q_\theta(x_l|\boldsymbol{x}_{<l}))\right] = \sum_{x_1, \ldots, x_L}\prod_{l=1}^L P_{<l}(x_l)\log(\gamma + (1-\gamma)Q_\theta(x_l|\boldsymbol{x}_{<l})) \tag{141}$$

$$= \sum_{l=1}^L \mathbb{E}_{\boldsymbol{x}_{<l} \sim P}\left[\sum_{x_l \in \mathcal{V}} P(x_l|\boldsymbol{x}_{<l})\log(\gamma + (1-\gamma)Q_\theta(x_l|\boldsymbol{x}_{<l}))\right]. \tag{142}$$

In others terms, this is equivalent to maximize the following loss for all $l \in \{1, \ldots, L\}$ and $\boldsymbol{x}_{<l} \sim P$, and denoting $q_l = Q_\theta(x_l|\boldsymbol{x}_{<l})$ and $p_l = P(x_l|\boldsymbol{x}_{<l})$:

$$l(\boldsymbol{q}) = \sum_{l=1}^L p_l \log(\gamma + (1-\gamma)q_l). \tag{143}$$

By using the Lagrange multiplier method, to ensure that the sum of the probabilities is equal to one, we have the optimization problem:

$$l(\boldsymbol{x}, \mu) = \sum_{l=1}^L p_l \log(\gamma + (1-\gamma)q_l) + \mu\left(1 - \sum_{l=1}^L q_l\right). \tag{144}$$

Using KKT conditions, we have:

$$\begin{cases} \forall l, \quad \nabla_{q_l} l(\boldsymbol{x}, \mu) = \frac{p_l(1-\gamma)}{\gamma + (1-\gamma)q_l} - \mu = 0, \\ \sum_{l=1}^L q_l = 1, \end{cases} \tag{145}$$

Which is equivalent to:

$$\forall l, \quad p_l = \mu\left(\frac{\gamma}{1-\gamma} + q_l\right) \tag{146}$$

By summing for all $l \in \{1, \ldots, L\}$, we have:

$$\mu = \frac{1-\gamma}{1 - \gamma + V\gamma}. \tag{147}$$

and thus, for all $l \in \{1, \ldots, L\}$:

$$q_l = \frac{p_l}{\mu} - \frac{\gamma}{1-\gamma} = \frac{p_l(1 - \gamma + V\gamma) - \gamma}{1 - \gamma}. \tag{148}$$

Finally, we can show that in that case:

$$q_l \lessgtr p_l \Leftrightarrow \frac{p_l(1 - \gamma + V\gamma) - \gamma}{1 - \gamma} \lessgtr p_l \tag{149}$$

$$\Leftrightarrow p_l(1 - \gamma + V\gamma) \lessgtr p_l(1 - \gamma) + \gamma \tag{150}$$

$$\Leftrightarrow p_l \lessgtr \frac{1}{V}, \tag{151}$$

which concludes the proof. $\qquad\square$

First, let us recall that precision $\alpha_\lambda$ can be written as a linear function of a divergence $\mathcal{D}_\lambda(P \parallel Q)$ denoted PR-Divergence in Verine et al. (2023):

$$\alpha_\lambda(P\|Q) = \min(\lambda, 1) - \mathcal{D}_\lambda(P \parallel Q). \tag{152}$$

It can be shown, using Lemma 4.4.1 in Verine (2024) that:

$$\mathcal{D}_\lambda(P \parallel Q) = \frac{1}{2} \sum_{\boldsymbol{x} \in \mathcal{V}^L} |\lambda P(\boldsymbol{x}) - Q(\boldsymbol{x})| - \frac{1}{2}|\lambda - 1|. \tag{153}$$

Therefore, using similar arguments as Ji et al. (2023), we can bound the divergence between $P$ and $Q$ by the divergence between the conditional distribution $P(\cdot \mid \boldsymbol{x}_{<t})$ and $Q(\cdot \mid \boldsymbol{x}_{<t})$:

$$\mathcal{D}_\lambda(P \parallel Q) = \frac{1}{2} \sum_{\boldsymbol{x} \in \mathcal{V}^L} |\lambda P(\boldsymbol{x}) - Q(\boldsymbol{x})| - \frac{1}{2}|\lambda - 1| \tag{154}$$

$$= \frac{1}{2} \sum_{x_1, \ldots, x_L} \left| \lambda \prod_{l=1}^{L} P(x_l \mid \boldsymbol{x}_{<l}) - \prod_{l=1}^{L} Q(x_l \mid \boldsymbol{x}_{<l}) \right| - \frac{1}{2}|\lambda - 1| \tag{155}$$

$$= \frac{1}{2} \sum_{x_1, \ldots, x_L} \left| \prod_{l=1}^{L} \left( \lambda^{\frac{1}{L}} P(x_l \mid \boldsymbol{x}_{<l}) \right) - \prod_{l=1}^{L} Q(x_l \mid \boldsymbol{x}_{<l}) \right| - \frac{1}{2}|\lambda - 1|. \tag{156}$$

Using the triangular inequality, it can be shown that for every $a_t, b_t \in \mathbb{R}$:

$$\left| \prod_{l=1}^{L} a_i - \prod_{l=1}^{L} b_i \right| \leq \sum_{l=1}^{L} |a_i - b_i| \times \left( \prod_{j=1}^{l-1} a_j \right) \times \left( \prod_{j=l+1}^{L} b_j \right). \tag{157}$$

Thus, by taking $a_i = \lambda^{\frac{1}{L}} P(x_l \mid \boldsymbol{x}_{<l})$ and $b_i = Q(x_l \mid \boldsymbol{x}_{<l})$, the PR-Divergence can be bounded by:

$$\frac{1}{2} \sum_{l=1}^{L} \sum_{x_1, \ldots, x_l} \prod_{j=1}^{l-1} \left( \lambda^{\frac{1}{L}} P(x_j \mid \boldsymbol{x}_{<j}) \right) \left| \lambda^{\frac{1}{L}} P(x_l \mid \boldsymbol{x}_{<l}) - Q(x_l \mid \boldsymbol{x}_{<l}) \right| \times \sum_{x_{l+1}, \ldots, x_L} \prod_{j=l+1}^{L} (Q(x_j \mid \boldsymbol{x}_{<j}))$$
$$- \frac{1}{2}|\lambda - 1|. \tag{158}$$

By marginalizing over the $x_{l+1}, \ldots, x_L$, the bound can be expressed as:

$$\frac{1}{2} \sum_{l=1}^{L} \lambda^{\frac{l-1}{L}} \sum_{x_l} \mathbb{E}_{\boldsymbol{x}_{<l} \sim P} \left| \lambda^{\frac{1}{L}} P(x_l \mid \boldsymbol{x}_{<l}) - Q(x_l \mid \boldsymbol{x}_{<l}) \right| - \frac{1}{2}|\lambda - 1|. \tag{159}$$

Finally, we can show that:

$$\mathcal{D}_\lambda(P \parallel Q) \leq \mathbb{E}_{\boldsymbol{x} \sim P} \left[ \sum_{l=1}^{L} \lambda^{\frac{l-1}{L}} \mathcal{D}_{\lambda^{\frac{1}{T}}} \left( P(\cdot \mid \boldsymbol{x}_{<l}) \parallel Q(\cdot \mid \boldsymbol{x}_{<l}) \right) \right] + \frac{1}{2} \left[ \left| \lambda^{\frac{1}{T}} - 1 \right| \sum_{l=1}^{L} \lambda^{\frac{l-1}{L}} - |\lambda - 1| \right]. \tag{160}$$

Now we need to compute:

$$\mathcal{D}_\lambda(P(\cdot \mid \boldsymbol{x}_{<l}) \parallel Q(\cdot \mid \boldsymbol{x}_{<l})) = \min(\lambda, 1) - \mathbb{E}_{x_l \sim P(\cdot|\boldsymbol{x}_{<l})}\left[\min\left(\lambda, \frac{Q_{<l}(x_l)}{P_{<l}(x_l)}\right)\right] \tag{161}$$

$$\tag{162}$$

Now the challenge is that we do not have access to the ground truth distribution $P$. Thus, we can sample from the $x_l$ from the true distribution $P_{<l}$ and approximate the expectation density using Definition 5.1:

$$P_{<l}(x_l) \approx \widehat{P}_{<l}^{x_l}(x_l) = \gamma \mathbb{1}_{\{x_l = x_l\}} + (1 - \gamma)\bar{Q}_{<l}(x_l), \tag{163}$$

where $\bar{Q}$ is the model distribution detached from the computation graph. In other words, $\nabla_\theta \bar{Q} = 0$. However, to estimate the expectation, we can either use a one-step Monte Carlo approximation similarly to Ji et al. (2023):

$$\mathbb{E}_{x_l \sim P(\cdot|\boldsymbol{x}_{<l})}\left[\min\left(\lambda, \frac{Q_{<l}(x_l)}{P_{<l}(x_l)}\right)\right] \approx \min\left(\lambda, \frac{Q_{<l}(x_l)}{\widehat{P}_{<l}^{x_l}(x_l)}\right) \tag{164}$$

$$= \min\left(\lambda, \frac{Q_{<l}(x_l)}{\gamma + (1 - \gamma)\bar{Q}_{<l}(x_l)}\right) \tag{165}$$

$$= \mathbb{1}_{\{\bar{Q}_{<l}(x_l)<\delta_\lambda\}}\frac{Q_{<l}(x_l)}{\gamma + (1 - \gamma)\bar{Q}_{<l}(x_l)} + \mathbb{1}_{\{\bar{Q}_{<l}(x_l)\geq\delta_\lambda\}}\lambda, \tag{166}$$

with $\delta_\lambda = \frac{\lambda\gamma}{1-(1-\gamma)\lambda}$. By differentiating the expectation, we can show that the gradient of the expectation is:

$$\nabla_\theta \mathbb{E}_{x_l \sim P(\cdot|\boldsymbol{x}_{<l})}\left[\min\left(\lambda, \frac{Q_{<l}(x_l)}{P_{<l}(x_l)}\right)\right] \approx \nabla_\theta \min\left(\lambda, \frac{Q_{<l}(x_l)}{\widehat{P}_{<l}^{x_l}(x_l)}\right) \tag{167}$$

$$= \mathbb{1}_{\{\bar{Q}_{<l}(x_l)<\delta_\lambda\}}\frac{\nabla_\theta Q_{<l}(x_l)}{\gamma + (1 - \gamma)\bar{Q}_{<l}(x_l)} \tag{168}$$

$$= \mathbb{1}_{\{\bar{Q}_{<l}(x_l)<\delta_\lambda\}}\frac{\bar{Q}_{<l}(x_l)\nabla_\theta \log Q_{<l}(x_l)}{\gamma + (1 - \gamma)\bar{Q}_{<l}(x_l)}. \tag{169}$$

Thus, using the Monte Carlo approximation and Assumption 5.1, we can show that minimizing the PR-Divergence is equivalent to maximizing the following loss:

$$\mathcal{L}_{\mathrm{PR}-\lambda-\mathrm{MC}}(\theta) = -\mathbb{E}_{\boldsymbol{x}\sim P}\left[\sum_{l=1}^{L}\lambda^{\frac{t-1}{T}}\mathbb{1}_{\{\bar{Q}_{<l}(x_l)\leq\delta_{\lambda^{1/T}}\}}\frac{\bar{Q}_\theta(x_l|\boldsymbol{x}_{<l})}{\gamma + (1 - \gamma)\bar{Q}_\theta(x_l|\boldsymbol{x}_{<l})}\log Q_\theta(x_l|\boldsymbol{x}_{<l})\right], \tag{170}$$

where $\delta_{\lambda^{1/T}} = \frac{\lambda^{1/T}\gamma}{1-(1-\gamma)\lambda^{1/T}}$.

# C. Training Algorithms details

In this section, we describe the training algorithm used to minimize a weighted Negative Log-Likelihood (NLL) loss. While the optimization loop remains the same across all methods, the weight computation $w(\boldsymbol{x}, l)$ varies depending on the chosen criterion: *Trunc*, *TruncR*, *c-Div*, or *λ-PR*.

The core idea is to compute importance weights based on a detached estimate of likelihood, denoted $\bar{Q}$. This value is treated as a constant during backpropagation, ensuring the weight computation does not interfere with gradient updates.

In Python, this is implemented using the `.detach()` method. For instance, if `logp = model(x)`, then `logp.detach()` returns a tensor with the same values but without gradient tracking. This means $\bar{Q}$ can be used for weighting without contributing to the gradient computation itself.

---

**Algorithm 1** Weighted NLL Training Algorithm

---

**Require:** Training dataset $\mathcal{D} = \{\boldsymbol{x}^{(i)}\}_{i=1}^N$, model $Q_\theta$, method method $\in \{$NLL, *Trunc*, *TruncR*, *c-Div*, λ-PR$\}$, hyperparameters $\Delta$, $K$ for *Trunc*, $\alpha$ for *c-Div*, $\gamma$ and $\lambda$ for $\lambda - PR$, learning rate $\eta$

 1: **for** each training step **do**
 2:    Sample a batch $\mathcal{B} \subset \mathcal{D}$
 3:    **if** method = NLL **then**
 4:       **for** each $\boldsymbol{x} \in \mathcal{B}$ and $l$ **do**
 5:          $w(\boldsymbol{x}, l) \leftarrow 1$
 6:       **end for**
 7:    **else if** method = Trunc or method = *TruncR* **then**
 8:       Compute log-likelihood $\bar{Q}(\boldsymbol{x})$ for all $\boldsymbol{x} \in \mathcal{B}$
 9:       Add $\bar{Q}(\boldsymbol{x})$ for all $\boldsymbol{x} \in \mathcal{B}$ to a rolling list $\mathcal{Q}$ of size $K$
10:       **if** method = *Trunc* **then**
11:          Compute threshold $\delta$ for the highest $\delta \times K$ values in $\mathcal{Q}$
12:          **for** each $\boldsymbol{x} \in \mathcal{B}$ and $l$ **do**
13:             $w(\boldsymbol{x}, l) \leftarrow \mathbb{1}_{\{\bar{Q}(\boldsymbol{x}) \geq \delta\}}$
14:          **end for**
15:       **else if** method = *TruncR* **then**
16:          Compute threshold $\delta$ for the lowest $\delta \times K$ values in $\mathcal{Q}$
17:          **for** each $\boldsymbol{x} \in \mathcal{B}$ and $l$ **do**
18:             $w(\boldsymbol{x}, l) \leftarrow \mathbb{1}_{\{\bar{Q}(\boldsymbol{x}) \leq \delta\}}$
19:          **end for**
20:       **end if**
21:    **else if** method = *c-Div* **then**
22:       **for** each $\boldsymbol{x} \in \mathcal{B}$ and $l$ **do**
23:          Compute $\bar{Q}_{<l}(x_l)$
24:          $w(\boldsymbol{x}, l) \leftarrow \bar{Q}_{<l}(x_l)^{1/2}$
25:       **end for**
26:    **else if** method = λ-PR **then**
27:       **for** each $\boldsymbol{x} \in \mathcal{B}$ and $l$ **do**
28:          Compute $\bar{Q}_{<l}(x_l)$
29:          $w(\boldsymbol{x}, l) \leftarrow \frac{\bar{Q}_{<l}(x_l)}{\gamma + (1-\gamma)\bar{Q}_{<l}(x_l)}$
30:       **end for**
31:    **end if**
32:    Compute gradient:
33:       $g \leftarrow \nabla_\theta \left( -\sum_{\boldsymbol{x} \in \mathcal{B}} \sum_{l=1}^L w(\boldsymbol{x}, l) \log Q_{<l}(x_l) \right)$
34:    Update parameters: $\theta \leftarrow \theta - \eta g$
35: **end for**

---

# D. Experiments

All experiments were conducted using Pytorch and HuggingFace Transformers. For MathQA-Python generation, we used vLLM library to speed up the generation process. We used both A100-80GB and H100-80GB GPUs for the experiments.

## D.1. Models and training details

**CodeContests.** We benchmarked Llama-3.1 8B/70B Instruct model before/after the RLEF. RLEF (Gehring et al., 2024) uses rule-based reward to fine-tune LLM with reinforcement learning. An interesting point, is that the increase of the Precision at the cost of the drop of the Recall after RLEF in Figure 4 echos the finding of Le Bronnec et al. (2024); Kirk et al. (2024) that RL training could reduce the diversity of the model output.

**Integer multiplication.** We trained a small replica of a Llama transformer model, with 4 hidden layers, an embedding dimension of 32, 4 attention heads and a feedforward dimension of 128. We trained the model using 25 000 samples. We used the Adam optimizer, with a learning rate of 0.001, a weight decay of 1, 500 epochs, and a batch size of 512 sequences.

**WritingPrompts & MathQA-Python.** We fine-tuned models based on the pre-trained Olmo-1B and Llama3.2-3B models. All models, regardless of the dataset and the loss function used, were trained for 3 epochs. For non-NLL losses, fine-tuning started from a checkpoint obtained after one epoch of NLL training. For all training, we used the Adam optimizer, with a constant learning rate of 1e-6, with 1000 linear warmup steps, a batch size of 8.

**Instruction tuning on Alpaca.** We fine-tuned Llama3.1-8B on the Alpaca dataset, to get a basic instruction-tuned model, capable of generalization. We use the same training setup as for the other datasets. For Alpaca generation, we used a reference temperature of 0.5 on most experiments, since we observed some degeneracies in the generation with a temperature of 1.0.

## D.2. Evaluation Methods

In the evaluation phase in the Section 6, we evaluate the model using different methods.

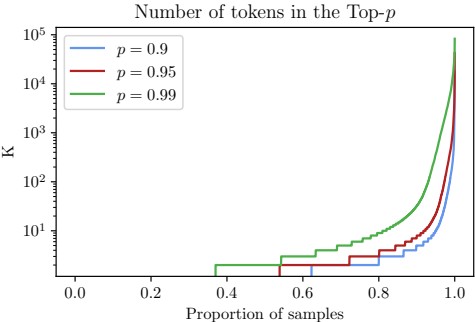
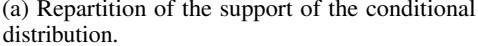
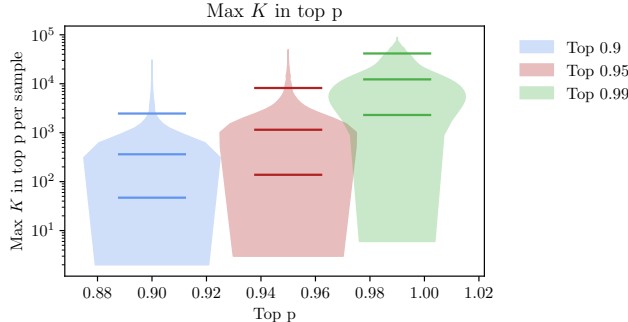

(a) Repartition of the support of the conditional distribution.

(b) Quantiles of the size of the support.

Figure 12: Support size of the conditional distribution $P(x_l \mid \boldsymbol{x}_{<l})$ estimated using a reference model Llama3-8B-Instruct on the CodeContests dataset. The left plot shows the distribution of the number of tokens needed to cover different mass $p$ (e.g., 0.9, 0.95, 0.99) for each position $l$. The right plot shows this numbers and the 10%, 50% and 90% quantiles.

**Estimating the Sparsity of $P$ via Token-Level Conditionals.** In this experiment, we aim to provide an empirical proxy for the sparsity of the true target distribution $P$. By quantifying how many tokens carry most of the probability mass under a strong reference model, we can validate the assumption that $P$ is indeed sparse in real-world settings. We approximate the sparsity of the target distribution $P$ using a reference model $Q$ assumed to approximate $P$ reasonably well. Specifically, we assume that $Q$ fits $P$ correctly in structure but remains noisy in estimation. This reflects realistic modeling conditions, where $Q$ captures the general shape of the target distribution but with limited precision due to model size or training noise. An

illustrative histogram of support sizes across examples is shown in Figure 12. We refer to this estimate as an upper bound on the true sparsity of $P$, as the method only considers local conditionals $Q(x_l \mid \boldsymbol{x}_{<l})$ and may include spurious high-entropy predictions not representative of the true support.

Regarding the use of the geometric mean to estimate sparsity: we rely on it because it is well suited to multiplicative or exponential behaviors, which naturally arise in token-level conditional probabilities in language models. At each position in a sequence, we compute the number of tokens needed to cover a fixed portion (e.g., 90%) of the total probability mass. Taking the maximum per sample, and then aggregating across the dataset using the geometric mean, allows us to capture a robust notion of sparsity that penalizes extremely large values less than the arithmetic mean would (which is desirable since these are rare), preserves scale-invariance (if all values are scaled by a constant factor, the geometric mean scales accordingly), reflects the multiplicative nature of uncertainty across positions in sequence models. This approach aligns with best practices in probabilistic modeling, such as those discussed in Murphy (2013) and other works on log-loss and information content.

---

**Algorithm 2** Estimating the Support Size of the Target Distribution

---

**Require:** Dataset $\mathcal{D} = \{\boldsymbol{x}^{(i)}\}_{i=1}^N$, reference model $Q_\theta$ (e.g., Llama3.1), coverage threshold $p \in (0, 1)$
1: **for** each sample $\boldsymbol{x} \in \mathcal{D}$ **do**
2:     **for** each position $l$ in the target sequence **do**
3:         Compute conditional distribution $Q_\theta(x_l \mid \boldsymbol{x}_{<l})$
4:         Sort vocabulary tokens by decreasing probability
5:         Compute minimal number $k_l$ of top tokens such that cumulative mass $\geq p$
6:     **end for**
7:     Let $k_{\max}(\boldsymbol{x}) \leftarrow \max_l k_l$
8: **end for**
9: Compute geometric mean of $\{k_{\max}(\boldsymbol{x})\}_{\boldsymbol{x} \in \mathcal{D}}$
10: **Return** geometric mean as upper bound estimate of $|\mathrm{Supp}(P)|$

---

**Precision and Recall for the Integers Multiplication Task.** In this experiment, we evaluate the model using the Precision and Recall metrics. We compute these metrics using the following algorithm:

---

**Algorithm 3** Precision and Recall for Integer Multiplication

---

**Require:** Number of samples $N$, model $Q_\theta$ trained to generate a sequence of digits $a_1 a_2 \times b_1 b_2 = c_1 c_2$, where $a_i, b_i, c_i \in \{0, 1, \ldots, 9\}$ such that $a_1 a_2 \times b_1 b_2 \% 97 = c_1 c_2$.
1: Initialize unique $\leftarrow \varnothing$
2: Initialize correct $\leftarrow 0$
3: **for** each sample $i = 1, \ldots, N$ **do**
4:     Sample a sequence $\boldsymbol{x}^{(i)}$ from the model
5:     **if** the sequence has the right format **then**
6:         **if** $a1 a_2 \times b_1 b_2 \% 97 = c_1 c_2$ **then**
7:             correct $\leftarrow$ correct $+ 1$
8:             **if** $(a_1 a_2, b_1 b_2) \notin$ unique **then**
9:                 unique $\leftarrow$ unique $\cup (a_1 a_2, b_1 b_2)$
10:             **end if**
11:         **end if**
12:     **end if**
13: **end for**
14: Compute Precision Precision $= \frac{\text{correct}}{N}$
15: Compute Recall Recall $= \frac{|\text{unique}|}{99 \times 99}$
16: **Return** Precision, Recall

---

**Precision and Recall for the WritingPrompts Task.** In this experiment, we evaluate the model using Precision and Recall, computed following the algorithm proposed by Le Bronnec et al. (2024), described below. We used the same

hyperparameter than the original paper, with two differences: instead of using the embedding of the last token to featurize texts, we averaged the embeddings, and we used 2,000 samples for the support estimation.

---

**Algorithm 4** Precision and Recall for WritingPrompts

---

**Require:** Number of samples $N = 2000$, model $Q_\theta$, dataset $\mathcal{D} = \{\boldsymbol{x}^{(i)}\}_{i=1}^N$, number of neighbors $k = 4$, embedding model $\phi$ = GPT2-Large.
1: Estimate the support of the distribution $P$ with $\mathcal{D}$ using $k$-NN based on Kynkäänniemi et al. (2019) and Le Bronnec et al. (2024) with embedding model $\phi$.
2: Initialize $\mathcal{Q} = \varnothing$
3: **while** $|\mathcal{Q}| < N$ **do**
4:     Sample a sequence $\boldsymbol{x}^{(i)}$ from the model and add it to $\mathcal{Q}$
5: **end while**
6: Estimate the support of the distribution $Q_\theta$ with $\mathcal{Q}$ using $k$-NN based on Kynkäänniemi et al. (2019) and Le Bronnec et al. (2024) with embedding model $\phi$.
7: Count the number of samples $N_{\text{fake}}$ of $\mathcal{Q}$ that are in $\operatorname{Supp} P$.
8: Count the number of samples $N_{\text{real}}$ of $\mathcal{D}$ that are in $\operatorname{Supp} Q_\theta$.
9: Compute Precision $= \frac{N_{\text{fake}}}{N}$
10: Compute Recall $= \frac{N_{\text{fake}}}{N_{\text{real}}}$
11: **Return** Precision, Recall

---

**Precision and Recall for the Code Generation Task and MathQA-Python.** We use the widely adopted **pass@k** metric to evaluate code-generation tasks in CodeContests and MathQA-Python. Formally, pass@k estimates the probability that at least one of $k$ sampled completions is correct, which directly corresponds to **Precision** when $k = 1$, and **Recall** as pass@100 − pass@1. This accounts for diversity in model outputs under multiple draws.

The computation uses the unbiased estimator for pass@k (Chen et al., 2021): given $n \geq k$ samples per prompt, let $c$ be the number of correct completions. The unbiased estimate is:

$$\widehat{\text{pass@}}k = \begin{cases} 1 - \frac{\binom{n-c}{k}}{\binom{n}{k}} & \text{if } c \leq n - k \\ 1 & \text{otherwise} \end{cases}$$

---

**Algorithm 5** Precision and Recall for Code Generation

---

**Require:** Dataset $\mathcal{D} = \{\boldsymbol{x}^{(i)}\}_{i=1}^N$, number of samples per prompt $n$, evaluation budget $k$
1: Initialize `sum_pass_1` $\leftarrow 0$
2: Initialize `sum_pass_k` $\leftarrow 0$
3: **for** each prompt $\boldsymbol{x}^{(i)} \in \mathcal{D}$ **do**
4:     Generate $n$ completions $\boldsymbol{y}^{(i,1)}, \dots, \boldsymbol{y}^{(i,n)}$ using $Q_\theta$
5:     Let $c \leftarrow$ number of correct completions among the $n$
6:     Compute unbiased estimate:
$$\widehat{\text{pass@}}k^{(i)} = \begin{cases} 1 - \frac{\binom{n-c}{k}}{\binom{n}{k}} & \text{if } c \leq n - k \\ 1 & \text{otherwise} \end{cases}$$
7:     `sum_pass_1` $\leftarrow$ `sum_pass_1` $+ c/n$
8:     `sum_pass_k` $\leftarrow$ `sum_pass_k` $+ \widehat{\text{pass@}}k^{(i)}$
9: **end for**
10: Precision $\leftarrow \frac{\texttt{sum\_pass\_1}}{N}$
11: Recall $\leftarrow \frac{\texttt{sum\_pass\_k}}{N} - \text{Precision}$
12: **Return** Precision, Recall

---

### D.3. Comparison with decoding methods.

To complement our analysis of temperature-based sampling, we report results for two alternative decoding strategies on the WRITINGPROMPTS dataset: Top-$p$ sampling, with varying $p$ and KL-guided temperature sampling (Chang et al., 2023), reported to yield better diversity in question-anwering and summarization tasks.

We tested Top-$p$ for $p \in \{0.1, 0.5, 0.8, 0.9\}$, and re-implemented KL-guided sampling, using guidance parameter $\sigma \in \{1.0, 3.0, 5.0, 10.0\}$ (same range as in the original paper). All experiments used the same model and evaluation setup as in the main paper.

Table 5: Precision (P) and Recall (R) on WRITINGPROMPTS for different decoding methods. Note that for as $\sigma$ increases, KL-guided approaches standard sampling. For all methods we used a temperature of $t = 1$.

| Method | P | R |
|---|---|---|
| Base sampling, $t = 1$ | 0.848 | 0.086 |
| Top-$p$, $p = 0.1$ | 0.997 | 0.001 |
| Top-$p$, $p = 0.5$ | 0.996 | 0.001 |
| Top-$p$, $p = 0.8$ | 0.886 | 0.033 |
| Top-$p$, $p = 0.9$ | 0.805 | 0.058 |
| KL, $\sigma = 1.0$ | 0.757 | 0.061 |
| KL, $\sigma = 3.0$ | 0.800 | 0.068 |
| KL, $\sigma = 5.0$ | 0.831 | 0.069 |
| KL, $\sigma = 10.0$ | 0.844 | 0.086 |

From Table 5, we observe that **Top-$p$ sampling increases Precision at the expense of Recall**. This aligns with intuition, as Top-$p$ discards low-probability tokens, favoring more likely completions and thus leading to higher Precision.

In contrast, KL-guided sampling reduces both Precision and Recall overall. This may be due to the fact that the method was originally designed for conditional generation tasks such as summarization and question answering, which likely have different characteristics than our open-ended generation setting.

These results reinforce the need for Recall-oriented training losses, which offer a more effective way to improve the Precision-Recall trade-off than decoding-based methods alone.

