# OpenReview forum: "Improving Diversity in Language Models: When Temperature Fails, Change the Loss"
_ICML.cc/2025/Conference — ICML 2025 poster_

### Official Review · Reviewer_2yG3 · 2025-03-12

**Overall Recommendation:** 3

**Summary:**

The paper investigates the impact of temperature scaling on the precision–recall (P&R) trade-off in language models. The authors provide a theoretical analysis showing that while lowering the temperature enhances precision, increasing it does not necessarily improve recall. They propose new loss functions (e.g., TruncR, c-Div, and λ-PR) to train models that emphasize recall, thereby allowing for a more balanced diversity–quality trade-off when temperature scaling is applied. Experimental evaluations on tasks such as code generation, integer multiplication, and writing prompts are used to validate the theoretical insights.

**Claims And Evidence:**

- The authors provide a theoretical investigation into how temperature affects P&R, providing analyses that explain the observed limitations of temperature scaling. However, Artificial case analysis is limited, as only a small set of cases are considered. Cannot give a general insight about the trends of P&R when varying temperature in general cases.

**Essential References Not Discussed:**

- Chang et al. KL-Divergence Guided Temperature Sampling.

- Lu et al. Diver: Large Language Model Decoding with Span-Level Mutual Information Verification

- Zhang et al. Trading Off Diversity and Quality in Natural Language Generation

- Chung et al. Increasing Diversity While Maintaining Accuracy: Text Data Generation with Large Language Models and Human Interventions

**Experimental Designs Or Analyses:**

- The experiment setup and evaluation criteria have some concerns (see comments).

**Methods And Evaluation Criteria:**

Strength
- Viewing the quality-diversity trade-off using precision and recall is interesting (however, it’s not new). The alternative loss functions to train models for higher recall seem to be effective. By shifting the focus from decoding adjustments to training objectives, the authors provide a new angle on tackling the quality–diversity trade-off.


Weakness
- While the claim in the main paper is general (change the loss function when the temperature sampling is based), there is a lack of comparison with existing decoding-based works that promote diversity along with quality. This is essential to understand the effectiveness compared to simpler methods (no need to train with a different loss function). One example is Chang et al. KL-Divergence Guided Temperature Sampling.

- The effects of changing the loss function on other tasks are not investigated. For example, does changing the loss function affect the generalization/in-context learning ability of LMs? This affects the utility of the proposed methods for general use.

**Other Comments Or Suggestions:**

- For the completeness of the paper, I suggest to discuss the detailed computation of Precision and Recall metrics for writing prompts experiment in the appendix.

- MAUVE and Average Pairwise Cosine Similarity are more common metrics to evaluate the quality and diversity of LLM responses. It would be better if the authors can evaluate the proposed methods on those metrics to give diverse insights.

**Other Strengths And Weaknesses:**

See comments and questions

**Questions For Authors:**

- Sentences 243 244 are vague to me and does not give enough context for the next drawback claim. What are specific trade-offs (is it about P&R only or about general measures? minimizing alternative f-divergences between what? Citations need for this sentence as well.)

- Why pass@100 - pass@1 measures the diversity of generated samples? if all generated samples are the most same with some different tokens. In this case, the pass@100 is high but diversity is low?

**Relation To Broader Scientific Literature:**

Most of the existing work on prompting diversity or balancing diversity-quality trade-off in LLMs is in the decoding phase. This paper brings new insight and claim that we should adjust the loss function to promote the effectiveness of temperature-scaling in decoding phase.

**Theoretical Claims:**

I have not closely verified the proofs.

---

> ### Author Rebuttal · Authors · 2025-03-30
>
> Thank you for the detailed review and your feedback.
>
> ## Decoding-based methods
>
> In our paper, we deliberately focused on the temperature parameter (commonly referred to as the "diversity" parameter) because its theoretical analysis already presented some complexity. For this reason, we chose to limit the scope of this study to ensure a focused exploration of the temperature parameter.
> However, we agree that comparing empirically with other decoding-based methods could bring other insights.
>
> As suggested, we investigated the effect of the following methods on the WritingPrompts dataset:
> - top-p (or nucleus) sampling
> - KL-Divergence Guided Temperature Sampling
>
> We re-implemented the KL-Guided sampling from scratch, since the original implementation was not compatible with our PyTorch models. We used the same range for $\sigma$ as in the original paper.
>
> |     | P     | R     |
> | --- | ----- | ----- |
> | NLL | 0.848 | 0.086 |
>
>
> #### Top-p
> | p   | Precision | Recall |
> | --- | --------- | ------ |
> | 0.1 | 0.997     | 0.001  |
> | 0.5 | 0.996     | 0.001  |
> | 0.8 | 0.886     | 0.033  |
> | 0.9 | 0.805     | 0.058  |
>
>
> #### KL-Guided
>
> | $\sigma$ | P     | R     |
> | -------- | ----- | ----- |
> | 1.0      | 0.757 | 0.061 |
> | 3.0      | 0.800 | 0.068 |
> | 5.0      | 0.831 | 0.069 |
> | 10.0     | 0.844 | 0.086 |
>
> (As $\sigma$ increases, KL-Guided approaches standard decoding.)
>
> We can conclude that Top-p increases Precision at the cost of Recall. Since Top-p removes the least probable tokens, we could intuitevely expect a higher Precision.
>
> KL-Guided seems to globally decrease both P&R. We believe that an explanation would be that this method has been mainly designed for conditional text generation, like summarization and question answering, which might exhibit specificites compared to our generation task.
>
> Overall, we believe that these results provide more grounding on the necessity of using Recall-oriented losses as opposed to simple decoding-based methods.
>
> ## General models
>
> To answer your question, we trained a general-purpose instruction model on the Alpaca dataset using our proposed losses, $\lambda$-PR and $c$-Div, to demonstrate their effectiveness on broader instruction tuning tasks. We then evaluated this model on both the WritingPrompts and MathQA-Python datasets.
>
> For MathQA-Python, which requires generating Python code from natural language questions, we used 3 in-context examples to prompt the model and evaluated it as in CodeContest.
> We used the same evaluation as for CodeContest. We chose MathQA over CodeContest because it offers more training data and is better suited to the capabilities of our models.
>
> ### Alpaca MathQA-Python
> | Method                                            | P     | R    |
> | -- | ----- | ---- |
> | NLL                                               | 0.088 | 0.39 |
> | c-Div ($c = 1.4$)                                 | 0.084 | 0.43 |
> | $\lambda$-PR ($\lambda = 0.75, \gamma = 1e^{-5}$) | 0.083 | 0.46 |
>
> On these results, we see that overall, our losses do not significantly impact Precision, meaning that generalization/in-context ability is not affected. However, we observe a significant increase in Recall, which is consistent with our previous findings.
>
> ### Alpaca WritingPrompts
>
> | Method                                            | P    | R     |
> | -- | --- | ----- |
> | NLL                                               | 0.83 | 0.040 |
> | c-Div ($c = 1.4$)                                 | 0.82 | 0.12  |
> | $\lambda$-PR ($\lambda = 0.75, \gamma = 1e^{-5}$) | 0.51 | 0.21  |
>
> On the WritingPrompts dataset, we observe that general models trained with our losses on the Alpaca dataset exhibit a similar pattern to the specialized models. We see a significant increase in Recall, sometimes at the expense of Precision.
>
> This suggests that our losses can consistently improve Recall even on more general models and tasks. We hope these additional experiments provide a more comprehensive view of the effectiveness of our proposed losses.
>
> ### Suggestions
> - We will add more details about P&R in the appendix, this should help clarify the evaluation process.
> - We computed MAUVE metrics for the WritingPrompts dataset. We will add it to Table 1.
>
> | Method  | MAUVE     |
> | -- | ----- |
> | Trunc   | 0.074 |
> | GOLD    | 0.005 |
> | Tailr   | 0.087 |
> | c-Div   | 0.068 |
> | Trunc-R | 0.073 |
> | λ-PR    | 0.096 |
> | NLL     | 0.104 |
>
>
> - For 243–244, we mean that image generation can be optimized for specific PR tradeoffs using f-divergences, as shown by Verine et al. (2023). However, these methods rely on assumptions that don’t hold for text due to its causal nature. We’ll clarify this in the final version.
> - In code generation, pass@100–pass@1 reflects the gain from sampling multiple candidates over one. In your example, the metric is relevant: if samples differ only slightly (e.g., variable names), structure diversity is low and pass@1 is likely high. We’ll clarify this with more details on the evaluation process.

---

> > ### Comment · Reviewer_2yG3 · 2025-04-03
> >
> > Thanks the authors for their very detailed responses and additional experiments. Since most of my questions were answered in the rebuttal, I updated the score accordingly.

---

### Official Review · Reviewer_2owJ · 2025-03-12

**Overall Recommendation:** 3

**Summary:**

This paper studies how recall and precision can be effectively traded off in language models. First, they study formal definitions of precision and recall in simplified settings and show cases where decreasing the temperature improves precision at the cost of recall, but increasing the temperature hurts both precision and recall. Motivated by the fact that it seems easier to improve precision than recall via temperature adjustment, they then propose recall-oriented loss functions. Empirically, (1) they confirm that it is difficult to improve recall by increasing temperature, and (2) they show that fine-tuning with recall-oriented loss functions and then decreasing temperature leads to a better precision-recall tradeoff than starting with a normally trained model and increasing temperature.

**Claims And Evidence:**

I really like the direction that the paper is going, as the claims are interesting and the problem is important. However, the experiments presented in the paper do not sufficiently support the claims, as discussed below. One other minor limitation of the paper is that I find the writing somewhat confusing (will provide concrete suggestions in a later section).

- The first main claim is that lowering temperature improves precision at the cost of recall, but increasing temperature typically harms both after a certain point. This claim is supported by experiments.

- The second main claim is that fine-tuning with recall-oriented loss and adjusting temperature attains a better precision-recall curve than doing normal NLL training and adjusting temperature. This claim is weakly supported by a single experiment on WritingPrompts. I find this experiment insufficient because:
  - (a) Because there is no "downstream" measure of precision and recall, they instead rely on an automatic measure based on a previously proposed method involving embedding the texts and measuring precision and recall in the embedded space. This evaluation setup provides useful evidence, but automatic metrics have been found to be flawed (see, e.g., [Gehrmann, Clark, and Sellam 2023](https://arxiv.org/abs/2202.06935)). Therefore, I think this claim needs to be evaluated with more metrics. One good one would be looking at pass@k on CodeContests, which the authors use as a dataset for other parts of the paper.
  - (b) I also think this claim needs to be evaluated on more than one model and more than one dataset.

- Minor point: section 6.2 claims that it "empirically confirms the theoretical insights from Theorem 4.2," but as far as I can tell it only measures support sparsity at various cutoffs, when Theorem 4.2 makes specific predictions about upper bounds for precision and recall measures. So the authors' claims that their experiments confirm their theory does not seem well-supported by evidence.

**Essential References Not Discussed:**

N/A

**Experimental Designs Or Analyses:**

- The paper provides some description of the experiments but does not describe them in enough detail for me to judge their soundness with confidence. Nonetheless, the experiments seem sound at first glance. Some details that I could not find:
  - How hyperparameters were chosen
  - Which model is used to produce embeddings for the automatic P&R metrics
  - Whether there was a train-test-val split

**Methods And Evaluation Criteria:**

The methods and datasets make sense for the problem.

**Other Comments Or Suggestions:**

Some writing suggestions:
- Overall, I think the paper could be streamlined in terms of making it clearer how the theory motivates the proposed solution, as well as omitting tangential results or making it clearer that they are tangential. For example, I did not find theorem 4.2 to be particularly compelling because as far as I understand, the bound is only useful for large $t$, but in practice people do not take temperature to be very far from 1.
- The second part of Section 4 is framed as using an artificial setting to make the prediction that very high temperature decreases both P&R. However, it's already well-known empirically that setting temperature very high is not effective. Nonetheless, I found the setting interesting because it could provide some intuition on the factors at play. I would suggest spending more time talking about the intuition behind the model, like how different factors affect the effect of temperature, rather than centering the section around the specific claim that high temperatures are harmful.
- I found it confusing to refer to the previous methods as "baselines" because they are not solving the same problem as this paper. I think it would be clearer to just refer to these methods as the precision-focused versions of the recall-focused losses in this work.

**Other Strengths And Weaknesses:**

N/A

**Questions For Authors:**

N/A

**Relation To Broader Scientific Literature:**

- The first contribution, studying how temperature affects precision and recall, seems well-studied. Nonetheless, the paper provides useful further experiments for this question.
- The second contribution, training LMs for recall to attain a better precision-recall tradeoff, seems new and valuable.
- Theory in section 4 (characterizing P&R tradeoffs in simplified settings): I found this analysis interesting but would characterize it more as a supplement and motivation for the other claims in the paper, rather than a standalone theoretical contribution.
- Theory in section 5 (characterizing the proposed recall-oriented losses): I see this theory as supplementing the main contribution of proposing recall-oriented training.

**Theoretical Claims:**

I read the theoretical claims in the main text and they seem reasonable. I did not check the proofs.

---

> ### Author Rebuttal · Authors · 2025-03-30
>
> We would like to thank Reviewer 2owJ for the detailed review and suggestions. We are glad you appreciate the paper.
>
> Following your suggestion, as well as those from other reviewers, we conducted additional experiments:
>
> **New dataset: MathQA-Python.**
> We trained a model on MathQA-Python and used the same evaluation as for CodeContest: Precision as pass@1, and Recall as the gain obtained from sampling (pass@100 - pass@1). We chose MathQA-Python over CodeContest because it offers more training data and is better suited to the capabilities of our models.
>
> **New model: Llama3.2-3b WritingPrompts.**
>   We trained an additional model on the WritingPrompts dataset.
>
> **General-purpose instruction model.**
>   We trained a general-purpose instruction model on the Alpaca dataset using our proposed losses, $\lambda$-PR and $c$-Div, to demonstrate their effectiveness on broader instruction tuning tasks and more expressive models. We then evaluated this model on both the WritingPrompts and MathQA-Python datasets.
>
> We report the results below and will update Table 1 in the final version. While time constraints limited the scope, we believe these additional experiments provide a strong indication the effectiveness of our losses on more tasks, models and metrics.
>
>
> ### Olmo1b MathQA-Python
>
> | Alpha                                            | P    | R    |
> | - | - | - |
> | NLL                                              | 0.42 | 0.36 |
> | c-Div (c=1.4)                                    | 0.30 | 0.46 |
> | $\lambda$-PR ($\lambda = 0.1, \gamma = 1e^{-7}$) | 0.06 | 0.48 |
> | TruncR  ($\Delta = 0.1$)                         | 0.29 | 0.43 |
>
> An interesting observation is that $\lambda$-PR impacts Precision more severely than the other losses, but ultimately achieves a high Recall. This suggests that the resulting model offers better coverage of the target distribution.
>
>
> ### Llama3.2-3b WritingPrompts
> | Method                                           | P    | R    |
> | - | - | - |
> | NLL (MLE)                                        | 0.77 | 0.08 |
> | c-Div ($c = 1.3$)                                | 0.72 | 0.17 |
> | $\lambda$-PR ($\lambda = 0.9, \gamma = 1e^{-5}$) | 0.59 | 0.19 |
>
> Note that for this larger model (compared to Olmo1b), we did not benchmark the TruncR loss due to its incompatibility with distributed training in the current implementation.
> We also observed that the model tuned with $\lambda$-PR was much more sensitive to the sampling temperature. We used $t=0.5$, as higher values led to some degeneracies. However, we could not investigate this further.
>
> ### Alpaca WritingPrompts
> | Method                                           | P    | R    |
> | - | ---- | ---- |
> | NLL                                              | 0.83 | 0.04 |
> | c-Div ($c = 1.4$)                                | 0.82 | 0.12 |
> | $\lambda$-PR ($\lambda = 0.5, \gamma = 1e^{-7}$) | 0.57 | 0.26 |
>
> ### Alpaca MathQA-Python
> | Method                                           | P    | R    |
> | -- | ---- | ---- |
> | NLL                                              | 0.09 | 0.39 |
> | c-Div ($c = 1.4$)                                | 0.08 | 0.43 |
> | $\lambda$-PR ($\lambda = 0.1, \gamma = 1e^{-5}$) | 0.08 | 0.42 |
>
> All Alpaca experiments used a temperature of $t=0.5$ to avoid degeneracies.
>
> ### Analysis
>
> We observe the same pattern as in the initial experiments. This confirms that our losses can achieve higher Recall than NLL, which we believe strengthens the claims and findings presented in the paper.
>
>
> ### Section 6.2
> - For Section 6.2, we will reformulate the text to clarify that the experiments are designed to verify the assumptions behind the theoretical analysis, not its results.
>
> ### Experiments
>
> - We used hyperparameters very similar to those described in the original Olmo paper (we only used a smaller learning rate of 1e-6 instead of 2e-6). To ensure comparability, we used the same optimization parameters for all losses.
> - For the PR metrics, we used the exact same setup as in the original paper (Le Bronnec et al., 2024), i.e., GPT2-large as the embedding model.
> - We trained all models under the same conditions (same number of epochs, same batch size, same optimizer, etc.) on the training set and reported the metrics on the validation set.
>
>
> We will incorporate these details in the appendix of the paper.
>
> ### Suggestions
> - We will gladly incorporate the suggested reformulations, it will indeed improve the overall flow of the paper. We could indeed move some parts to the appendix and spend more time discussing the idea behind the model.
>
> For a side note, in Theorem 4.2, the bounds rely on *almost* no assumption on the target distribution or the model, and the bound is not necessarily useful for large $t$, especially if $Z$ is low. For instance, when the model is uncertain (i.e., $Z$ is low), the bound may still be useful even for $t < 1$. But we agree that this is more a general theoretical result than a practical one.

---

> > ### Comment · Reviewer_2owJ · 2025-04-04
> >
> > Thanks for the detailed rebuttal!
> >
> > The additional experiments look promising but only validate one half of the claim, which is that recall-oriented training improves recall at the cost of precision. But the more important claim to me is that lowering the temperature then leads to a better precision-recall tradeoff compared to adjusting the temperature of NLL. Do you have experiments showing this claim on the additional settings?

---

> > > ### Author Response · Authors · 2025-04-07
> > >
> > > Thank you for your answer! We acknowledge that our earlier reference to a "better tradeoff" should be clarified in the light of these new experiments. To clarify, our new experiments illustrate two distinct improvements achieved by our proposed losses compared to standard NLL:
> > >
> > > - At the highest Recall achievable by NLL (optimized via temperature tuning), our method consistently attains superior Recall at the same Precision level. This improvement is demonstrated in our experiments on MathQA with Olmo1B, in the table below (and also supported in the initial experiments in Figure 5, comparing the 70B model with the 70B-RLEF variant, although the RLEF variant itself was not trained using our proposed losses, RLEF is a Precision-oriented model). This supports our assertion that for use-cases prioritizing Recall, our losses provide an improved Precision-Recall tradeoff.
> > > - We identify scenarios in which our approach consistently achieves higher Recall across the entire spectrum of Precision levels attainable by NLL. This is demonstrated in the new experimental results with Llama 8B trained on Alpaca, summarized in the table below (and supported in the paper on Fig. 7 and 8.)
> > >
> > > We will add these new results in the next revision.
> > >
> > > ## Olmo MathQA-Python
> > > **Highest Recall of NLL, R=0.47:**
> > > | Method                                            | P    | R     |
> > > | ------------------------------------------------- | ---- | ----- |
> > > | NLL (temperature=1.6)                                               | 0.20 | 0.47 |
> > > | c-Div ($c = 1.4$, temperature=1.0)                                 | 0.21 | 0.50  |
> > >
> > > ## Alpaca MathQA-Python
> > >
> > > **Highest Recall of NLL, R=0.49:**
> > > | Method                                            | P    | R     |
> > > | ------------------------------------------------- | ---- | ----- |
> > > | NLL (temperature=1.0)                                               | 0.067 | 0.49 |
> > > | c-Div ($c = 1.4$, temperature=0.8)                                 | 0.087 | 0.49  |
> > >
> > > **Highest Precision of NLL, P=0.10:**
> > > | Method                                            | P    | R     |
> > > | ------------------------------------------------- | ---- | ----- |
> > > | NLL (temperature=0.1)                                               | 0.10 | 0.10 |
> > > | c-Div ($c = 1.4$, temperature=0.1)                                 | 0.10 | 0.14  |

---

### Official Review · Reviewer_DUE2 · 2025-03-14

**Overall Recommendation:** 3

**Summary:**

The paper provides a detailed analysis of the relationship between temperature, precision, and recall, offering insights into why lowering the temperature improves quality (precision), while increasing the temperature usually does not enhance coverage (recall). The paper primarily addresses two key questions: the impact of temperature adjustment on the precision-recall trade-off in language models, and how to train models to improve recall. By proposing recall-oriented loss functions, it presents a method to achieve a better P&R trade-off through temperature scaling, and validates this approach experimentally. The main contributions of the paper are as follows:

1. Analysis of the impact of temperature scaling on the P&R trade-off
2. Proposal of recall-oriented loss functions
3. Experimental validation of theoretical findings

**Claims And Evidence:**

The author’s experiments and theoretical demonstrations extensively validate their claims:

1. The analysis of the impact of temperature on P&R is thorough and detailed, and the improvements to the loss function for the P&R trade-off have achieved the stated effects.
2. The experimental section sufficiently validates the relevant theoretical findings through three different scenarios.

**Essential References Not Discussed:**

I haven't found it yet.

**Experimental Designs Or Analyses:**

The experimental design by the authors is adequate, but some additional experiments might be necessary:

1. The authors provide three scenarios to answer the three proposed questions, yet in the subsequent analysis, not all of these questions seem to be fully addressed by the three given scenarios. Therefore, additional experiments may be needed to ensure that the issues claimed in the three scenarios are adequately answered (e.g., in Section 6.4, only the WritingPrompts dataset task is discussed).
2. The analysis of the experiments in Section 6 is somewhat confusing, and the appearance of some figures is not clearly related to the problems they aim to address.

**Methods And Evaluation Criteria:**

1. The method designed by the author analyzes the relevant impact of temperature on P&R, and the method design is reasonable.
2. The author compares a sufficient number of baselines.

**Other Comments Or Suggestions:**

Some images appear in the main text but are not mentioned in the text.

**Other Strengths And Weaknesses:**

I haven't found it yet.

**Questions For Authors:**

Please refer to the above parts.

**Relation To Broader Scientific Literature:**

I haven't found it yet.

**Theoretical Claims:**

The theoretical proof results in the paper are clear and well-presented, with complete and detailed proofs provided, although I did not check all the details.

---

> ### Author Rebuttal · Authors · 2025-03-30
>
> We thank Reviewer DUE2 for the review and constructive feedback. We have added further experiments with more datasets and models; please refer to our response to Reviewer 2owJ for details. While the first question has only one relevant scenario, we believe the second and third questions require multiple scenarios to answer meaningfully.
>
> **The authors provide three scenarios to answer the three proposed questions, yet in the subsequent analysis, not all of these questions seem to be fully addressed by the three given scenarios. Therefore, additional experiments may be needed to ensure that the issues claimed in the three scenarios are adequately answered (e.g., in Section 6.4, only the WritingPrompts dataset task is discussed).**
>
>
> Following your suggestion, as well as those from other reviewers, we conducted additional experiments within the limited available time to further investigate the effect of our Recall-oriented losses.  (Similar response as to Reviewer 2owJ)
> The new experiments include:
>
> - **New dataset: MathQA-Python.**
>   We trained a model on the MathQA-Python dataset. Using the same proxies for evaluating Precision and Recall as in CodeContest.
>
> - **New model: Llama3.2-3b WritingPrompts.**
>   We trained an additional model on the WritingPrompts dataset.
>
> - **General-purpose instruction model.**
>   We trained a general-purpose instruction model on the Alpaca dataset using our proposed losses, $\lambda$-PR and $c$-Div, to demonstrate their effectiveness on broader instruction tuning tasks. We then evaluated this model on both the WritingPrompts and MathQA-Python datasets.
>
> We report the results in the tables below and will extend Table 1 accordingly in the final version of the paper.
>
>
> ### Olmo1b MathQA-Python
>
> | Alpha                                            | P    | R    |
> | ------------------------------------------------ | ---- | ---- |
> | NLL                                              | 0.42 | 0.36 |
> | c-Div (c=1.4)                                    | 0.30 | 0.46 |
> | $\lambda$-PR ($\lambda = 0.1, \gamma = 1e^{-7}$) | 0.06 | 0.48 |
> | TruncR  ($\Delta = 0.1$)                         | 0.29 | 0.43 |
>
> An interesting observation is that $\lambda$-PR impacts Precision more severely than the other losses, but ultimately achieves a high Recall. This suggests that the resulting model offers better coverage of the target distribution.
>
>
> ### Llama3.2-3b WritingPrompts
> | Method                                           | P    | R    |
> | ------------------------------------------------ | ---- | ---- |
> | NLL (MLE)                                        | 0.77 | 0.08 |
> | c-Div ($c = 1.3$)                                | 0.72 | 0.17 |
> | $\lambda$-PR ($\lambda = 0.9, \gamma = 1e^{-5}$) | 0.59 | 0.19 |
>
> Note that for this model, which is larger than Olmo1b, we did not benchmark the TruncR loss, as the current implementation is not compatible with distributed training.
> We also observed that the model tuned with $\lambda$-PR was much more sensitive to the sampling temperature. We used $t=0.5$, as higher values led to some degeneracies. However, due to limited time, we could not investigate this further.
>
> ### Alpaca WritingPrompts
> | Method                                           | P    | R    |
> | ------------------------------------------------ | ---- | ---- |
> | NLL                                              | 0.83 | 0.04 |
> | c-Div ($c = 1.4$)                                | 0.82 | 0.12 |
> | $\lambda$-PR ($\lambda = 0.5, \gamma = 1e^{-7}$) | 0.57 | 0.26 |
>
> ### Alpaca MathQA-Python
> | Method                                           | P    | R    |
> | ------------------------------------------------ | ---- | ---- |
> | NLL                                              | 0.09 | 0.39 |
> | c-Div ($c = 1.4$)                                | 0.08 | 0.43 |
> | $\lambda$-PR ($\lambda = 0.1, \gamma = 1e^{-5}$) | 0.08 | 0.42 |
>
> We started from a pre-trained Llama3.1-8B model and used the same training setup as described in the paper. This yields a basic instruction-tuned model, capable of generalization (but still with limited capacity compared to SOTA models).
> Note that for all experiments conducted on Alpaca, we used a temperature of $t=0.5$ to avoid degeneracies. As with previous experiments, we could not benchmark the TruncR loss due to its incompatibility with distributed training.
>
> ### Analysis
>
> We observe the same pattern as in the initial experiments. This confirms that our losses can achieve higher Recall than NLL, which we believe strengthens the claims and findings presented in the paper.

---

> > ### Comment · Reviewer_DUE2 · 2025-04-03
> >
> > Thanks to the author for the detailed response. I will maintain my rating.

---

### Official Review · Reviewer_pQ4a · 2025-03-18

**Overall Recommendation:** 4

**Summary:**

Increasing diversity in language models requires careful tuning of decoding temperature. This paper shows that lowering temperature improves precision, but raising it often fails to enhance recall and effective tunability demands training models focusing on coverage. This paper provides two settings where the precision would fail to improve and recall can also go down provably. Then it provides a series of loss functions that can be used to perform fine-tuning on LLMs that claim to improve the recall. Then results are provided for three tasks, for all the models considered and proposed loss functions.



## update after rebuttal

I've updated my ratings reflecting my satisfaction with the rebuttal.

**Claims And Evidence:**

The mentioned proofs and assumptions are correct. The evidence is clear and convincing and the claims aren't problematic. I do have certain questions on how certain quantities are computed, which I've deferred for the later section.

**Essential References Not Discussed:**

N/A

**Experimental Designs Or Analyses:**

The experimental aspect seems correct. However, the presentation for some of the graphics can be improved to make it widely appreciable by people with visual impairment.

I think more datasets can be used where it is often desirable to generate multiple solutions and then used in conjunction with a verifier (see -- Generative Verifiers: Reward Modeling as Next-Token Prediction). Such datasets may be -- MATH, AIME 2024.

**Methods And Evaluation Criteria:**

Need more datasets, such as MATH, AIME2024, MathQA-Python.

**Other Comments Or Suggestions:**

N/A

**Other Strengths And Weaknesses:**

Following are my questions --

1. I am not fully able to understand how the support is computed in practice, or how M can be computed. Why can one find an upper bound on the cardinality of support set by the geometric mean of the $\max_{\ell}\mathcal{S}(x_{\ell})$?? Shouldn't cardinality be monotonic with $\ell$ and with that how does this max operation over $\ell$ not have a trivial solution?
2. How is the top-p region estimated? The writing is not very clear, and I am not sure what is being done algorithmically. I'd appreciate an algorithmic block for every computation done in the paper.
3. For the toy example, it would be good to remind the readers that the actual PR is computed over the entire distribution till length L and not on the conditionals. I got very confused initially since I was thinking about conditional distribution PR, and it was clear only when I read the proof.
4. Do all mentioned conditions to have a strictly increasing recall have to hold simultaneously? Moreover, what happens when $\rho \approx 1$ , that is, Q is very close to P?
5. I am quite a bit confused about the turnc loss function, and its practical implementation aspect. For the starters, what is $\bar{Q}$? When we are training, initially the model I am assuming should have a lower $\delta$ but over time $\delta$ should increase, right?
6. Can authors add proof to Proposition 5.2? Below the same prop, why is a sampling from Q difficult? Moreover, if someone swaps the P and Q, isn't it somewhat similar to PPO, assuming the log-likelihood under the true distribution is the reward model (which we can replace with a teacher model's likelihood function)? This makes me feel that there should be a natural baseline like this in the work (formulating and using RLHF to improve recall)
7. Equation 18 seems non-differentiable as it has a parameter inside an indicator function. This needs clarification and again an algorithmic block.
8. Why is $\lambda$-PR separately plotted (which increases as y=x line) as another method to improve PR in Fig4? Isn't it the case that all the PR plots when varying temperatures are at fixed \lambda? What is even the use case of having a plot between $\alpha_{\lambda}$ and $\beta_{\lambda}$ if truly one always varies $\lambda$ in the entire range [0, $\infty$]
9. For Fig 6, what is the $\lambda$, otherwise if it is plotted for all lambda then it kind of conflates the trend with different hyperparameters or loss functions.
10. Which of the considered tasks have the sparsest distribution?

**Questions For Authors:**

See weakness section.

**Relation To Broader Scientific Literature:**

Improving on PR is an important task, and increasing temperature is something people think can lead to nice and diverse solutions. Therefore, I think this paper is a good contribution to a variety of sub-community in LLM, including synthetic data generation to verification.

**Theoretical Claims:**

The theoretical claims are correct. I've read through the appendix and examined the two artificial cases closely. I think assumptions are reasonable, and the two cases span the most practical settings. Overall I like the math aspect quite a lot.

---

> ### Author Rebuttal · Authors · 2025-03-30
>
> We thank Reviewer pQ4a for the thoughtful feedback and careful review. We are grateful for the time and effort.
> First, regarding the algorithmic questions raised by several reviewers: we will include detailed algorithm blocks for all computations in the appendix.
>
> We address explicit questions below and will incorporate all clarity improvements in the next revision.
>
> **Additional experiments**
>
> Following your suggestion, we added experiments on the MathQA-Python dataset (details in response to reviewer DUE2). We also trained a general-purpose instruction model on the Alpaca dataset, and observed a similar trend as in our initial experiments. We hope these results further support our method's generalizability.
>
>
>
> **Computing the support**
> - In practice, the support size of $P$ is indeed intractable. However, we can still obtain a rough characterization of the sparsity of $P$ (introduced in Thm 4.2) by leveraging the sparsity of the token-level conditional distributions $P(\cdot \mid x_{<l})$. To this end, we approximate the reference $P$ using a strong model $Q_{\theta}$ (Llama3.1-8B in our experiments). For each sample and each token position $i$ in the solution, we compute the conditional distribution $Q_{\theta}(\cdot \mid x_{<i}) \in \mathbb{R}^V$, where $V$ is the vocabulary size. At each position, we then determine the smallest number of tokens $n_{\mathrm{top_p}}$ that account for $p \in {0.9, 0.95, 0.99}$ of the total probability mass. For each sample, we take the maximum of these $n_{\mathrm{top_p}}$ values across all positions, and finally, we compute the geometric mean of these maxima across the dataset to estimate the sparsity of $P$.
>
> - We refer to this estimate as an "upper bound" since the true sparsity of $P$ is likely lower than what we obtain from the conditional distributions alone.
>
> - We will make sure to include this explanation in the final version of the paper.
>
> **Confusing explanations**
> - Thanks for the feedback, we will better distinguish between PR over the full distribution and conditional PR.
>
> **Conditions for an increasing Recall**
> - Yes, all conditions mentioned in Prop. 4.3 should hold simultaneously. In the case $\rho \approx 1$, the temperature does not help, as the model is already close to the target distribution.
>
> **Eq.18 and TruncR loss.**
> - We indeed differentiate only the terms outside the indicator function. The notation $\bar{Q}$ indicates that we use the value of $Q$ in the implementation, but no gradient flows through it (achieved via `detach()` in PyTorch). In Eq. (18), this implies that the gradient of the sum is zero when the condition is not satisfied. We will add an algorithm block to clarify this in the final version.
> - Regarding $\delta$, it characterizes the proportion of samples to keep. So, yes your interpretation is correct. During training, samples below the $\delta$ threshold will have their likelihood increased. To maintain a fixed proportion of selected samples, $\delta$ will gradually increase.
>
> **Prop 5.2 + teacher model**
> - Proposition 5.2 is proved in Appendix B.1, but is not clearly referenced in the main part. We will fix this. By "difficult," we refer to the fact that both of both sampling from $Q$ and computing $P(\cdot | x_{<l})$, since $P$ is unknown.
> - In a different setup, you're right that $P$ could be approximated by a teacher model. But in that case, the objective would be to match the Recall of the target model to that of the teacher, not to the true distribution. This, along with the idea of using RLHF to improve Recall, are very interesting directions for future work, but we believe they deviate somewhat from the main focus of our paper and theoretical analysis. We will nonetheless add a discussion on this in the final version of the paper.
>
> **Fig 4 PR**
> - Unlike Figures 5, 6, and 7, which display empirical values of Precision and Recall (computed using the metrics from Le Bronnec et al., 2024), Figure 4 is a theoretical illustration of the effect of each loss on the PR-curve, as in Verine et al., 2023. It is parametrized by $\lambda$, and showing both $\alpha_\lambda$ and $\beta_\lambda$ is standard in that context. Our goal was to illustrate that optimizing the $\lambda$-PR objective increases the corresponding tradeoff value for a given $\lambda_0$. In particular, for $\lambda_0 < 1$ (i.e., below the line $y = x$), Recall increases more than Precision.
>
> **Fig 6 $\lambda$**
> - The three points corresponds to models trained with $\lambda$-PR for $\lambda=1$ (matching TailR loss), and some $\lambda<1$. We observe that $\lambda < 1$, Precision decreases while Recall increases, as expected.
>
> **Tasks sparsity**
> - In addition to CodeContest, we evaluated the sparsity of WritingPrompts (plotted in the table below), a creative task. The sparsity is lower than for CodeContest, which could be expected as this is a less constrained task.
>
> Top-p  | 0.9   | 0.95  | 0.99
> - | - | - | - |
> |Supp(P)\|/V | 3.86% | 7.80% | 24.9%

---

> > ### Comment · Reviewer_pQ4a · 2025-04-04
> >
> > Thanks for the added clarifications. Can you also clarify why is geometric mean is a good strategy?
> >
> > I am also raising scores.

---

> > > ### Author Response · Authors · 2025-04-07
> > >
> > > We thank Reviewer pQ4a for the constructive feedback and for the updated score.
> > >
> > > Regarding the use of the geometric mean to estimate sparsity: we rely on it because it is well suited to multiplicative or exponential behaviors, which naturally arise in token-level conditional probabilities in language models.
> > >
> > > In our context, at each position in a sequence, we compute the number of tokens needed to cover a fixed portion (e.g., 90%) of the total probability mass. Taking the maximum per sample, and then aggregating across the dataset using the geometric mean, allows us to capture a robust notion of sparsity that:
> > > - Penalizes extremely large values less than the arithmetic mean would (which is desirable since these are rare),
> > > - Preserves scale-invariance (if all values are scaled by a constant factor, the geometric mean scales accordingly),
> > > - Reflects the multiplicative nature of uncertainty across positions in sequence models.
> > >
> > > This is consistent with best practices in probabilistic modeling and has also been discussed in sources such as Murphy (2012) and other works on log-loss and information content. We will make sure to add this clarification in the paper.

---

### Decision · Program_Chairs · 2025-05-01

**Decision:**

Accept (poster)

**Comment:**

The paper addresses the challenge of improving diversity in language models by analyzing the limitations of adjusting decoding temperature, demonstrating theoretically and empirically why increased temperature often fails to enhance recall. The authors propose alternative recall-oriented loss functions (e.g., TruncR, c-Div, λ-PR) that, when combined with temperature scaling, yield better precision-recall trade-offs. Extensive theoretical analyses are supported by comprehensive experimental validation across multiple datasets, demonstrating generalizability and effectiveness.

Strengths:
- Theoretical insights clearly justify the inadequacy of traditional temperature scaling in enhancing recall.
- Novel loss functions provide a robust and effective method to achieve improved recall without severely compromising precision.
- Comprehensive empirical validation across diverse scenarios, models, and tasks strengthens the paper’s findings.

Weaknesses:
- Initial clarity issues regarding the computation of certain metrics and theoretical formulations, although largely resolved in rebuttal.
- Early experimental results somewhat limited, subsequently strengthened significantly through rebuttal phase additions.

Overall, the paper contributes valuable theoretical and empirical insights into a fundamental problem in language modeling. After incorporating additional clarifications and experiments, reviewers uniformly raised their scores, emphasizing the significance and thoroughness of the contributions. Therefore, this paper is recommended for acceptance.